# Spatial analysis of the glioblastoma proteome reveals specific molecular signatures and markers of survival

Marie Duhamel [1,7] ✉, Lauranne Drelich[1,7], Maxence Wisztorski [1,7], Soulaimane Aboulouard [1], Jean-Pascal Gimeno[1], Nina Ogrinc [1], Patrick Devos [2], Tristan Cardon [1], Michael Weller [3], Fabienne Escande [4], Fahed Zairi[5], Claude-Alain Maurage[4], Émilie Le Rhun [1,3,4,8] ✉, Isabelle Fournier [1,6,8] ✉ & Michel Salzet [1,6,8] ✉

Molecular heterogeneity is a key feature of glioblastoma that impedes patient stratification and leads to large discrepancies in mean patient survival. Here, we analyze a cohort of 96 glioblastoma patients with survival ranging from a few months to over 4 years. 46 tumors are analyzed by mass spectrometry-based spatially-resolved proteomics guided by mass spectrometry imaging. Integration of protein expression and clinical information highlights three molecular groups associated with immune, neurogenesis, and tumorigenesis signatures with high intra-tumoral heterogeneity. Furthermore, a set of proteins originating from reference and alternative ORFs is found to be statistically significant based on patient survival times. Among these proteins, a 5-protein signature is associated with survival. The expression of these 5 proteins is validated by immunofluorescence on an additional cohort of 50 patients. Overall, our work characterizes distinct molecular regions within glioblastoma tissues based on protein expression, which may help guide glioblastoma prognosis and improve current glioblastoma classification.

Glioblastoma represents the main malignant primary brain tumor[1]. Their prognosis is poor with a median survival estimated at 16 months in clinical studies[2–7] and around 12 months in contemporary population-based studies[8]. Approximately 5% of patients survive more than 5 years[1]. Favorable therapy-independent prognostic factors include younger age and higher neurological performance status at diagnosis. Furthermore, low postoperative residual tumor volume has been associated with improved outcome. In a cohort of 232 patients with centrally confirmed glioblastoma who survived at least 5 years, the median age at diagnosis was 52 years

(range 21–77 years), and most patients had a gross total resection initially[9].

Morphological criteria for the diagnosis of glioblastoma, according to the World Health Organization (WHO) central nervous system tumor classification of 2021[10] include mitotic activity, anaplastic nuclear features, microvascular proliferation, and necrosis. Morphological variants include giant cell glioblastoma, gliosarcoma, and epithelioid glioblastoma. Isocitrate dehydrogenase (*IDH*) 1 or 2 mutations now exclude the diagnosis of glioblastoma. Tumors with morphological features of glioblastoma which exhibit *IDH* mutations are now

[1]Univ.Lille, Inserm, CHU Lille, U1192, Laboratoire Protéomique, Réponse Inflammatoire et Spectrométrie de Masse (PRISM), F-59000 Lille, France. [2]Univ. Lille, CHU Lille, ULR 2694 - METRICS: Évaluation des technologies de santé et des pratiques médicales, F-59000 Lille, France. [3]Department of Neurology & Clinical Neuroscience Center, University Hospital and University of Zurich, Zurich, Switzerland. [4]CHU Lille, Service de biochimie et biologie moléculaire, CHU Lille, F-59000 Lille, France. [5]CHU Lille, Service de neurochirurgie, F-59000 Lille, France. [6]Institut Universitaire de France (IUF), 75000 Paris, France. [7]These authors contributed equally: Marie Duhamel, Lauranne Drelich, Maxence Wisztorski. [8]These authors jointly supervised this work: Émilie Le Rhun, Isabelle Fournier, Michel Salzet. ✉e-mail: marie.duhamel@univ-lille.fr; emilie.lerhun@usz.ch; isabelle.fournier@univ-lille.fr; michel.salzet@univ-lille.fr

referred to as Astrocytoma, *IDH*-mutant, WHO grade 4[11]. Conversely, *IDH* wild-type tumors that do not fulfill morphological WHO grade 4 criteria are still diagnosed as glioblastoma if they exhibit at least one of the following alterations: EGFR amplification, a +7/−10 genotype or TERT promoter mutation[11]. Standard treatment of glioblastoma includes maximum safe resection followed by radiotherapy with concomitant and maintenance temozolomide[7].

Efforts to further subclassify glioblastoma have been restricted to the genomic, transcriptomic, and epigenetic levels. In 2008, the Cancer Genome Atlas (TCGA) group delineated three main signaling pathways affected by genetic alterations in glioblastoma, receptor tyrosine kinase/*RAS/PI3K*, *p53*, and *RB*[12]. Genome methylation profiling in adult patients with IDH wild-type glioblastoma allowed the definition of three epigenetic subtypes, (i) receptor tyrosine kinase (*RTK*) I often with *PDGFR* amplification, (ii) *RTK II* or classical often with *EGFR* amplification, *CDKN2A/B* deletion, and *PTEN* mutation, and (iii) mesenchymal[13]. Any clinical relevance of the methylation classes in glioblastoma remains controversial. The DNA methylation-based classification of CNS tumors has meanwhile evolved to a comprehensive machine-learning approach[14] that has shaped the new WHO classification[10], resulting also in the delineation of further rare methylation classes of glioblastoma. Prior to the introduction of methylation profiling, a classification based on transcriptional profiling revealed four subtypes of glioblastoma: proneural, neural, classic, and mesenchymal[15]. The neural subtype is no longer maintained since it may reflect contamination by normal brain tissue, but it has become apparent that transcriptomic profiles are less homogeneous and stable than genome or methylome classifiers. Despite these efforts, these approached have found limited clinical application, and only a few biomarkers are being used in clinics. Proteomic approaches have been less frequently explored, although they can identify and quantify the final product of altered genomics and transcriptomics and may better characterize the activation of specific pathways[16–18]. Proteomic analyses of gliomas have been performed to identify proteomic differences between grades and genomic alterations[19]. More recently, proteogenomic approaches have been used to stratify glioblastoma patients demonstrating a stronger association of protein expression with patient survival compared to RNA transcripts[20]. Another study has performed a multi-omics strategy to investigate glioblastoma biology[21]. However, glioblastoma are highly heterogeneous tumors, and a spatially resolved proteomics approach may bring new insights in glioblastoma biology to improve their stratification. The determination of specific proteomic signatures could help to improve the distinction between the different glioblastoma subtypes and guide the management.

In this study, we present a spatially resolved proteomic approach to characterize glioblastoma. We have analyzed a cohort of 96 glioblastoma patients of varying survival. A spatially resolved proteomic approach guided by mass spectrometry imaging enables us to stratify patients into three molecular groups. Based on this proteomic approach, we identify five prognostic protein markers. The expression of these five proteins are indicators of short and long survival and can therefore help to stratify patients. We validate our results on an external cohort of 50 glioblastoma patients by immunofluorescence. Altogether, these results highlight the potential of spatially resolved proteomics to decipher glioblastoma molecular heterogeneity and to identify markers associated with survival.

## Results
### Clinical characteristics
Fifty glioblastoma samples from a prospective cohort were collected (Supplementary Fig. 1A). Four tumors with an IDH1 mutation were excluded from the study (Supplementary Fig. 1A, tumor samples with a star). Among the remaining 46 patients (Table 1), 31 (67%) were male, the median age at diagnosis was 60 (interquartile range (IQR), 51–66),

**Table 1 | Clinical characteristics**

| | Total population (*n* = 46) |
|---|---|
| **Sex** | |
| Female, *n* (%) | 15 (33) |
| Male, *n* (%) | 31 (67) |
| **Age at diagnosis (years)** | |
| Median (IQR) | 60 (51–66) |
| **Karnofsky's performance status at diagnosis** | |
| Median (IQ) | 90 (80–90) |
| 0–80, *n* (%) | 14 (30) |
| 90–100, *n* (%) | 32 (70) |
| **Main location of the tumor** | |
| Frontal, *n* (%) | 11 (24) |
| Occipital, *n* (%) | 3 (6) |
| Parietal, *n* (%) | 12 (26) |
| Temporal, *n* (%) | 20 (43) |
| **Extent of surgical resection** | |
| Gross total, *n* (%) | 26 (57) |
| Partial, *n* (%) | 19 (41) |
| Biopsy, *n* (%) | 1 (2) |
| **MGMT promoter methylation status** | |
| Not methylated, *n* (%) | 31 (67) |
| Methylated, *n* (%) | 15 (33) |
| **EGFR amplification** | |
| No, *n* (%) | 22 (48) |
| Yes, *n* (%) | 24 (52) |
| **Chromosome 7 gain combined with chromosome 10 loss (+7/−10)** | |
| No, *n* (%) | 12 (26) |
| Yes, *n* (%) | 34 (74) |
| **EGFR amplification combined with 7 gain/10 loss** | |
| EGFR amplification or gain 7/lost 10 | 41 (89) |
| EGFR amplification without gain 7/lost 10 | 7 (15) |
| EGFR amplification and gain 7/lost 10 | 17 (37) |
| Gain 7/lost 10 without EGFR amplification | 17 (37) |
| **Homozygous CDKN2A deletion** | |
| No, *n* (%) | 18 (39) |
| Yes, *n* (%) | 28 (61) |
| **Median follow-up (months)** | |
| Median (IQR) | 19.4 (13.5–32.0) |
| **Initial treatment** | |
| RT/TMZ followed by six cycles of TMZ, *n* (%) | 18 (39) |
| RT/TMZ followed by then more than six cycles of months TMZ, *n* (%) | 4 (9) |
| RT/TMZ followed by less than six cycles of TMZ, *n* (%) | 20 (43) |
| Other treatment*, *n* (%) | 2 (4) |
| Clinical study, *n* (%) | 1 (2) |
| No treatment, *n* (%) | 1 (2) |
| **Progression** | |
| Yes, *n* (%) | 38 (83) |
| No, *n* (%) | 3 (6) |
| Unknown, *n* (%) | 5 (11) |
| **Progression-free survival (months)** | |
| Median (IQR) | 10.6 (7.1–16.3) |
| **Treatment at first progression (*n* = 38)** | |
| Yes, *n* (%) | 33* (87) |
| No, *n* (%) | 5 (13) |
| **Death** | |
| Yes, *n* (%) | 43 (93) |
| No, *n* (%) | 3 (7) |
| **Survival from surgery (months)** | |
| Median (IQR) | 19.4 (13.5–32.0) |
| **Survival** | |
| Upper IQR, *n* (%) | 12 (26) |
| Intermediate IQR, *n* (%) | 23 (50) |
| Lower IQR, *n* (%) | 11 (24) |

*IQR* interquartile range, *MGMT* O⁶-methylguanine DNA methyltransferase, *RT* radiotherapy, *SRT* stereotactic radiotherapy, *TMZ* temozolomide.
*One patient: RT only, one patient: six cycles TMZ then SRT.

the median Karnofsky performance status at diagnosis was 90 (80–90). Twenty-six (57%) patients had a gross total resection. A methylation of the MGMT promoter was noted in 15 tumors (33%), an *EGFR* amplification was noted in 24 cases (52%) and a homozygous *CDKN2A* deletion in 34 cases (74%). A standard treatment was initiated in 42 patients (91%). At the time of the analysis, 38 patients (83%) had progressed. After a median follow-up of 19.4 months (IQR 13.5–32), 43 patients (93%) had died. The median overall survival was 19.4 months. The pathologist defined regions of interest for each tumor sample: tumor, very dense and dense infiltration, necrosis, and microvascular proliferation (MVP), after hematoxylin-eosin Safran (HES) staining (Supplementary Fig. 1B).

## MALDI-MSI allows patient grouping based on molecular features

Considering the heterogeneity of glioblastoma, we conducted spatially resolved proteomic studies guided by mass spectrometry imaging (MSI) on the 46 tumors from the prospective cohort of patients (Fig. 1A). The MALDI-MSI was performed after on-tissue tryptic digestion and protein distribution in the MALDI images was therefore obtained based on their tryptic digestion peptides. Different proteins are found by MALDI-MSI in the different histological regions annotated by the pathologist (all tissue annotations provided in Supplementary Figs. 1B and 2). Several hundreds to thousand signals are detected for each single image corresponding to different tryptic peptides associated to different proteins. Figure 1B provides a few examples of proteins with different distribution in the different regions of the samples (extracted from cases 7, 14 and 41). On case 7, tryptic peptide showing a *m/z* of 1443.08 is a marker of necrosis regions and shows a complementary distribution to *m/z* 1199.12 that is specific to the tumor region. Similarly for case 14, the digestion peptide with *m/z* 1106.98 is only distributed in the necrosis region while the *m/z* 1802.15 is specific to the tumor and the *m/z* 1786.25 is specific of the presence of microvascular proliferations (MVP) in the tumor region. On case 41, four different proteins with distinct *m/z* were selected and plotted on the same image, demonstrating the specificity of the different proteins according to the different regions of the tissues and their different histopathological features. To grasp all the molecular changes associated with all the detected molecules and to get the main molecular discriminative feature, we then performed an individual first (Fig. 1C and Supplementary Fig. 2), and then a global segmentation of the MALDI-MSI data (Fig. 1D). Segmentation is a non-supervised multivariate statistical analysis which classify the image pixels according to the similarities/differences of their MS spectra. Since the studied cohort is only based on glioblastoma patients, this approach will highlight the different tumor subtypes since molecules always present in the different regions will not be discriminative between the patients unlike was obtained from the distribution of the different markers without the use of any statistical approach (Fig. 1B). Indeed, after individual segmentation the comparison between the histological annotations and the MSI molecular images shows less direct correlation (Fig. 1C and Supplementary Fig. 2). Interestingly, in some cases, necrotic regions or regions with MVP identified by the pathologist were not discriminated anymore after segmentation, indicating that other molecular changes were weighting more on the classification of the pixels associated with these regions, which we could hypothesized to be driven by patients belonging to mixt proteomic subtypes. To find how patients would stratify within the cohort we then considered the global segmentation (Fig. 1D). Three main regions were identified i.e. red (region A), yellow (region B), and blue (region C) according to the segmentation map considering the main molecular differences (Fig. 1D). Interestingly, common molecular features and discriminative ones were found in the cohort. Overall, three regions were found which were not anymore assigned to a unique histological region but were composed of several microenvironments depicting the different

proteomic glioblastoma subtypes (Supplementary Table 1). Each colored region shared common molecular characteristics, meaning that the spectra in each of these areas were similar. Some specific ions can be attributed to each region: *m/z* 967.62 and 1492.92 were specifically present in the region A, *m/z* 1914.59, 2375.07, and 2376.27 were specific to the region B, and *m/z* 1473.31, 2045.82, 2046.62, and 2237.85 were specific to the region C. Images of some group-specific ions are shown in Supplementary Fig. 1C, D. The Ward clustering method using IMAGEREVEAL MS Ver.1.1 software confirmed the segmentation of the 46 tumors into three groups with similar specific ions (Supplementary Fig. 1D, E).

In order to validate the classification obtained by MALDI-MSI, we analyzed 30 samples by SpiderMass[22,23] which is an ambient MS technology. SpiderMass provides the measurement of the lipids and metabolites present in the tissues. The MS spectra acquired in the positive ion mode from the different areas of 30 tumors tissues were submitted to a PCA analysis to find how the data were grouping. The features of the PCA were then subjected to a supervised analysis using linear discriminant analysis (LDA)[24,25] where the three groups were well separated based on their lipid content (Fig. 1Ea). According to Fig. 1Ea, LDA 1 discriminated region A from region C and the LDA 2 separated region B from regions A and C. The LDA analysis of the SpiderMass data therefore allowed the samples to be grouped in the same way as the MALDI-MSI classification. Some examples of discriminant ions (*m/z*) between the three regions, corresponding to lipids, are presented as their normalized intensities in Fig. 1Eb. The most discriminating peaks for group A in LD2 + correspond to *m/z* 746.75, and 810.65; for group B in LD2- correspond to *m/z* 718.55, 724.65, 744.55, 751.55, 778.55, 862.65, and 890.65; for group C in LD1- correspond to *m/z* 725.4, 754.6, 788.65, and 936.85. To consolidate the classification, validation was performed using either 20% randomly patients taken out or the one-patient-out method (Supplementary Table 1). Excellent cross-validation results were obtained using 20% randomly patient-taken-out method with 100% and 91.85% correct classification rates with and without outliers, respectively, and good classification using the one-patient-out method with 92.92% and 77.78% including or not outliers, respectively (Supplementary Table 1). These results of outliers and misclassifications (mainly group B) reflect the fact that each group is not represented by only one colored region.

## Identification of specific signaling pathway signatures for each group

To understand the molecular differences between the three regions, spatially resolved tissue proteomic was undertaken on the 46 tissue samples. On each tissue, 2–5 specific microextraction points were selected according to the molecular regions identified by spatial segmentation of MALDI-MSI data (Supplementary Data 1) to analyze the tumor heterogeneity and microenvironment presenting with specific protein signatures in each group. This resulted in a total of 135 microextraction points. Each extraction point was associated with one of the three regions identified by MALDI-MSI (red-A, yellow-B, and blue-C). In all tumor samples, 28 extractions were performed in the red region (A), 20 in the yellow region (B) and 87 in the blue region (C) (Supplementary Tables 2 and 3 and Supplementary Fig. 1B). These three molecular regions are histologically heterogeneous (Supplementary Table 2). In the yellow region, 45% of the micro-extracted points were in the tumor, 25% in the tumor including few necrotic cells, 5% in the tumor with MVP and 20% in the necrotic area. In the red region, 39.3% of the micro-extracted points were in the tumor, 14.3% in the very dense infiltration area, 28.6% in the dense infiltration area, 10.7% in the tumor including few necrotic cells, 3.6% in the tumor with MVP, and 3.6% in tumor with inflammation. In the blue region, 73.6% of the micro-extracted points were in the tumor, 5.7% and 2.3% in the very dense and dense infiltration areas respectively, 3.4% in the tumor including few necrotic cells, 12.6% in the tumor with MVP, 1.1% in the

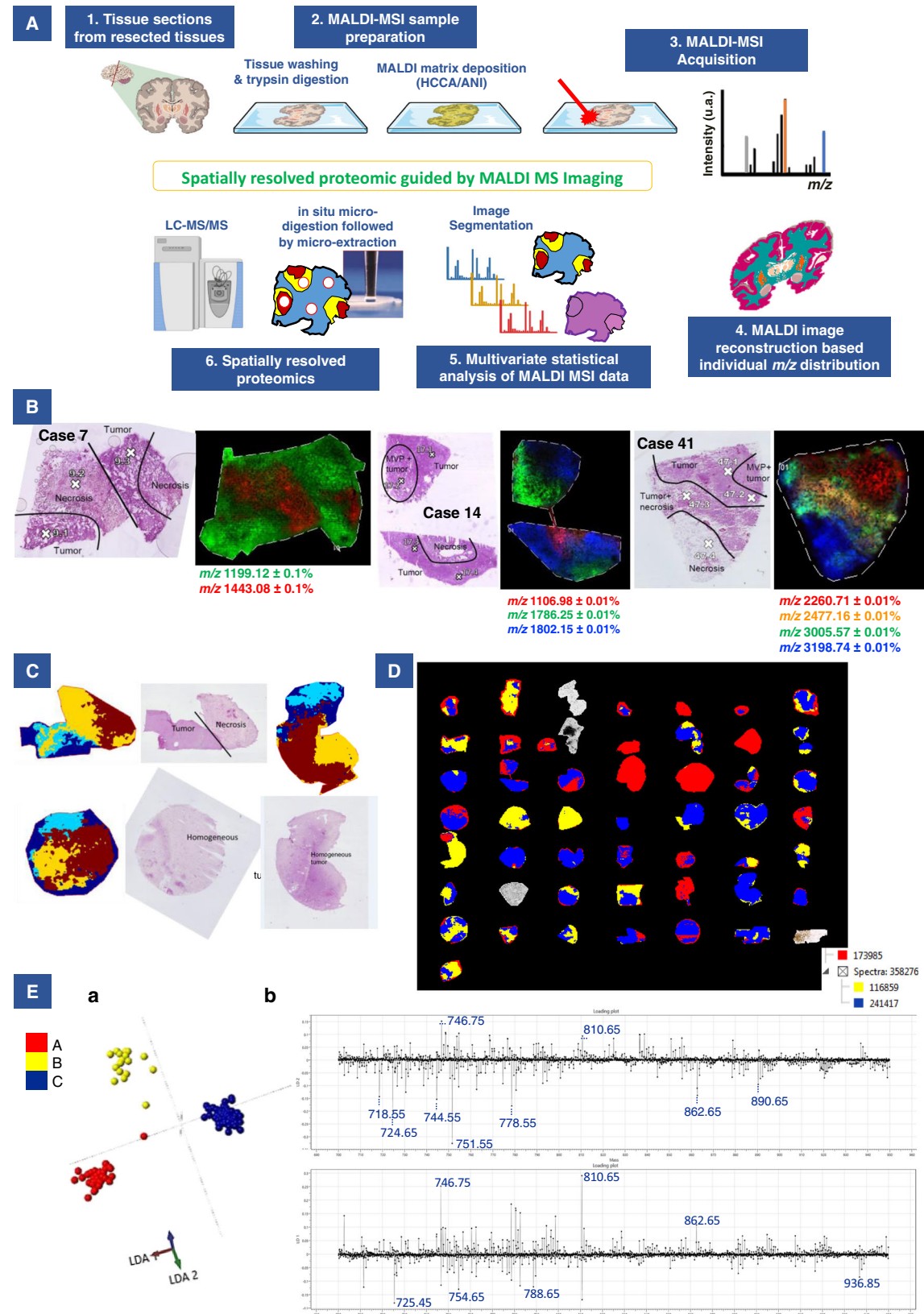

tumor with inflammation and 1.1% in the necrotic region (Supplementary Table 2).

From these shotgun spatial proteomic experiments, a total of 4936 proteins were identified (Supplementary Data 1). First, we measured the correlation between all the extraction points from the 46 glioblastoma samples by a Pearson correlation analysis. This analysis allows the grouping of the samples according to their similarities in the relative quantification of the identified proteins, thus in an unsupervised approach without bias, and not considering the grouping obtained by the MALDI-MSI. Hierarchical clustering of all the samples based on the correlation coefficients between them reveals a grouping of the samples like the MSI-identified colored regions (Fig. 2A). The

**Fig. 1 | Histological, MALDI-MSI, and SpiderMass data. A** General workflow of the MALDI-MS imaging combined with microproteomics used for glioblastoma inter- and intratumor heterogeneities characterization (Created with BioRender.com). **B** Selection of specific *m/z* ions identified by MALDI-MSI showing a correlation with histological regions annotated by the pathologist. One ion is represented by one color. **C** Representative annotated histopathology images of three glioblastoma samples and their corresponding segmentation map obtained from MALDI-MSI data. Colors represent molecularly different regions. Note that for two different tissues, similar colors are not equivalent to similar molecular groups. The segmentation map shows different clusters for each case and non-observable with HES coloration. Complete annotations for all samples are provided in Supplementary Fig. 2. **D** Global segmentation maps of all tissues together after MALDI-MSI analysis.

Colors represent molecularly different regions as shown in the corresponding dendrogram. The segmentation map gives 3 main clusters. The four tumors which are not segmented correspond to the IDH-mutant tumors, which were excluded from the analysis. **E** The built PCA-LDA classification model based on three glioma groups: Group A (red), Group B (yellow), and Group C (blue). a LDA representation of the 3-class PCA-LDA (right). The table (right) represents the "20% out" and "leave-one-patient-out" cross-validation results of the built classification model. b LD2 loading spectra (top) indicate the discrimination between Group A (red) and Group B (yellow). The ten most discriminatory lipid peaks are indicated by the blue dash line. LD1 loading spectra (bottom) indicate the discrimination between Group A (red) and Group C (blue). The ten most discriminatory lipid peaks are indicated by the blue dash line.

first cluster regroups mainly samples belonging to the red region. The second cluster contains only samples belonging to the yellow region while the third cluster is more represented by samples extracted from the blue region. With this analysis, we confirmed the existence of distinct proteomics glioblastoma subtypes cross-validating the results issued from the MALDI-MSI segmentation. To better understand the differences between each identified group, ANOVA tests with a Benjamini Hochberg FDR of 0.05 was performed. A total of 1183 proteins showed a significant difference in expression between the three groups (Fig. 2B and Supplementary Data 2). Two main branches were identified in the heatmap. The first branch was composed of 100% of samples extracted from the yellow region (region B). The second branch separates group A (red region) from group C (blue region). This branch is then separated into two sub-branches with the first one corresponding to region A and regrouping 79.2% of the samples extracted from the red region and 20.8% of the samples extracted from the blue region. The second sub-branch corresponds to the largest cluster, group C and contains 82.8% of the samples extracted from the blue region, 9% of the samples extracted from the red region and 8% of the samples extracted from the yellow region. We confirmed that each sample from the same-colored region has the same proteomic profile (Supplementary Data 2). Three specific clusters of overexpressed proteins for each region were identified using the heatmap (Fig. 2B) i.e., cluster 1 corresponds to proteins overexpressed in group B; cluster 2, to proteins overexpressed in group A and cluster 3, to proteins overexpressed in group C. The lists of overexpressed proteins per group are presented in Supplementary Data 2.

In group A (mainly represented in cluster 2), the proteins are associated with neuro-developmental genes, that are characteristic of neuronal/glial lineages or progenitor cells. Most proteins were related to neurogenesis and axon guidance (dihydropyrimidinase-related protein 1 (CRMP1), misshapen-like kinase 1 (MINK1), neuromodulin (GAP43), dihydropyrimidinase-related protein 5 (DPYSL5), dihydropyrimidinase-related protein 4 (DPYSL4), microtubule-associated protein tau (MAPT), kinesin-like protein KIF2A (KIF2A), neurofilament heavy polypeptide (NEFH), unconventional myosin-XVIIIa (MYO18A), MAGUK p55 subfamily member 2 (MPP2), alpha-internexin (INA), CLIP-associating protein 2 (CLASP2) (Supplementary Data 3). Using the functional enrichments analysis tool of String database, the most representative Reactome pathway was devoted to axon guidance. Nine of the 16 proteins identified in this pathway are involved in neuron development projection, morphogenesis, and guidance (Supplementary Fig. 3Aa). System biology analyses using SNEA and Cytoscape confirmed that the proteins in group A (Cluster 2) are involved in neurite outgrowth, synaptogenesis, synaptic vesicle transport and neurotransmission (Fig. 2C). Interestingly, among the identified proteins some are known to be involved in tumorigenesis like mitogen-activated protein kinase 3 (MAPK3), protein kinase C alpha type (PRKCA) and some were already identified in glioblastoma e.g. CRMP1, DPYSL2 (i.e. CRMP2)[26], DPYSL5 (i.e. CRMP5)[27], GAP43[28–30], as well as Tau protein encoded by MAPT in low-grade glioma[31].

Proteins overexpressed in group B (mainly represented in cluster 1) were linked to microglial activation and more generally immune system activation. Indeed, among the proteins identified, 10 proteins are linked to the immune response such as complement C1q subcomponent subunit C and B (C1QC and C1QB), complement factor H (CFH), haptoglobin (HP), kininogen-1 (KNG1), histidine-rich glycoprotein (HRG), transthyretin (TTR), grancalcin (GCA), proteins S100A9 (S100A9) & S100A12 (S100A12), erythrocyte band 7 integral membrane protein (STOM) and galectin-3-binding protein (LGALS3). Immunoglobulin heavy and light chains (IGHG2; IGKC; IGHG1; IGLC6; IGHM and IGHA1) and macrophage markers, macrophage-capping protein (CAPG) were also detected (Supplementary Data 3). Moreover, some proteins are related to iron transporters like ceruloplasmin (CP), serotransferrin (TF), hemopexin (HPX) and haptoglobin, and other proteins are associated to coagulation e.g., transthyretin, kininogen-1 (KNG1), plasminogen (PLG). Most of these proteins are known to be present in human plasma. These results are in accordance with histological annotations reflecting that most of the extraction points belonging to region B present intense proliferation of capillary endothelial cells with inflammation and hemorrhage (Supplementary Fig. 1B). The cytoscape and SNEA analysis (Fig. 2D) confirmed that most of the proteins are involved in the complements and coagulation cascades, inflammation, ischemia, vascularization, wood healing, and cancer. The same pathways were found in Reactome (Supplementary Fig. 3Ab). Some of these proteins have already been identified in the TCGA glioma database (see below) and are mostly associated with unfavorable prognosis, e.g., Grancalcin and CAPG (Supplementary Fig. 3B). These results are in accordance with histological annotations reflecting that most of the extraction points of the region B are in areas of intense proliferation of capillary endothelial cells with inflammation and hemorrhage (Supplementary Fig. 1B).

The overexpressed proteins in group C (mainly represented in cluster 3) are mainly involved in tumor growth (Hepatoma-derived growth factor (HDGF), Developmentally regulated GTP-binding protein 2 (DRG2)), but also in virus infection (Eukaryotic translation initiation factor 3 subunit L (EIF3L), Double-stranded RNA-binding protein Staufen homolog 1 (STAU1) and Interferon-induced double-stranded RNA-activated protein kinase (EIF2AK2)) (Supplementary Data 3). KEGGS analyses confirmed a network of proteins involved in Epstein–Barr virus infection (Supplementary Fig. 3Ac). Cytoscape pathway analyses established that this group is linked to viral infection and antiviral immune response (Fig. 2D). System biology analyses confirmed the involvement of proteins in virus infection (transfection, reproduction) and transcriptomic modification at the RNA level (RNA splicing, metabolism, replication) (Fig. 2D). Some other markers of the group C are known to be bad prognosis indicators such as EIF2AK2 and ZC3HAV1.

## Identification of alternative proteins

Using the OpenProt alternative proteins database[32], 257 AltProts were identified in our glioblastoma cohort and 170 were quantified in addition to the previously described proteins issued from the

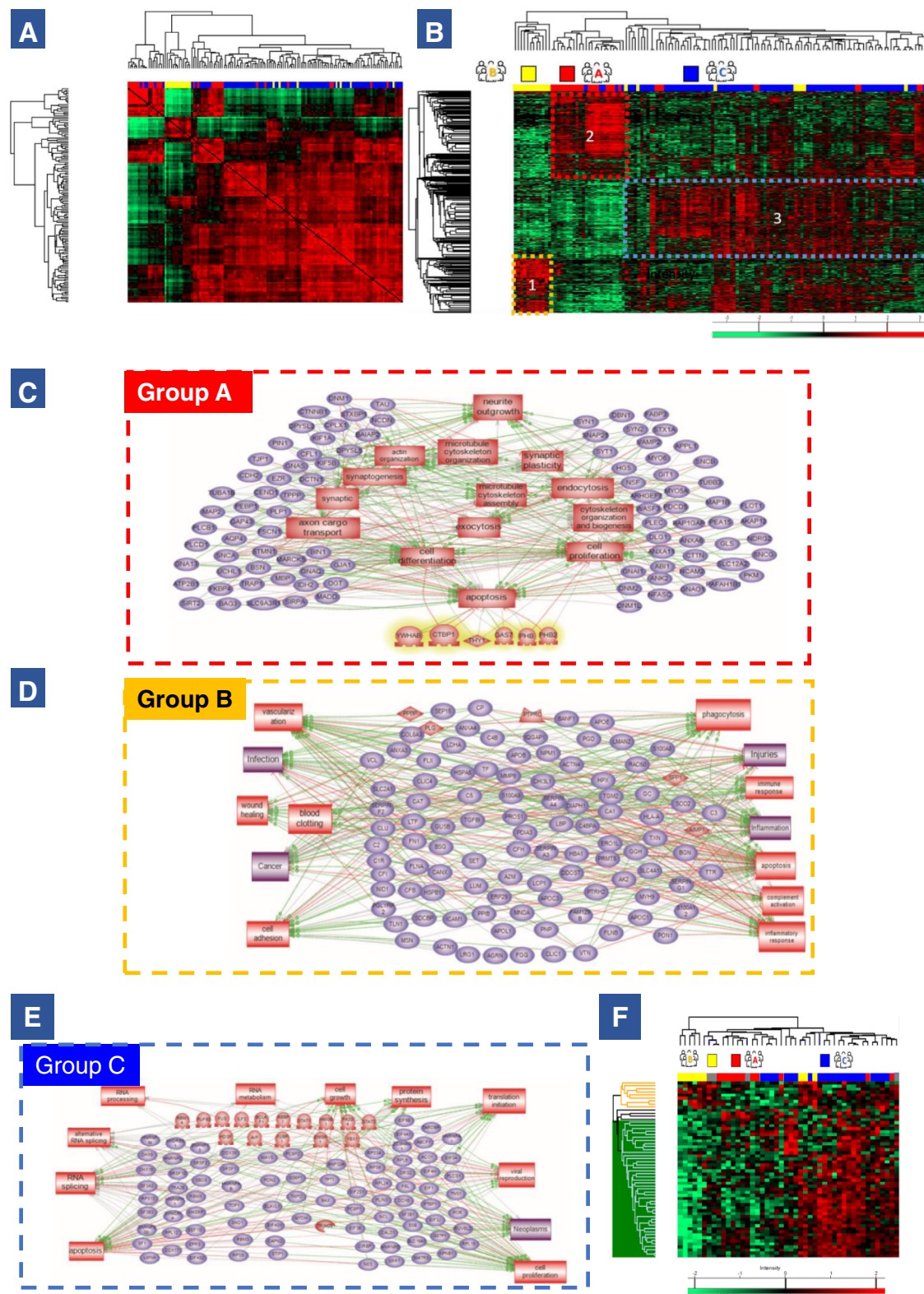

**Fig. 2 | Spatially resolved shotgun proteomics analysis. A** Matrix correlation map between all microextraction points from the 46 tumors. Correlation coefficients are calculated between each sample and are represented on a heatmap. **B** Heatmap of proteins with different regulation profiles as determined after label-free quantification in the three groups highlighting the presence of three clusters. Shotgun proteomics was performed after on-tissue trypsin digestion followed by micro-extraction at the spots determined from MALDI-MSI data. **C** Pathway analysis of proteins overexpressed in group A reveals that a large majority of protein is involved in (a) neurogenesis, brain development, synaptogenesis and cytoskeleton organization. **D** Pathway analysis of proteins overexpressed in group B reveals that majority of proteins are involved in injuries, inflammation, and more generally immune system response and vascularization. **E** Pathway analysis of proteins overexpressed in group C shows implication in cell proliferation, neoplastic processes, RNA metabolism and processing and viral reproduction. **F** Heatmap of alternative proteins with different regulation profiles as determined after label-free quantification in the three regions highlighting the presence of three clusters. Source data are provided as a Supplementary Data file.

conventional ORF known as the reference proteins (RefProts). After ANOVA tests with a $P$ value of 0.05, 58 were differentially expressed between the three regions (Fig. 2F). In region A, four AltProts are overexpressed coming from ncRNA, IP_2390879 issued from *LOC107985743*, IP_244732 from *KIFC3*, involved in cell adhesion, IP_672223 from *GBP1P1* and IP_710015 from *LRRC37A9P* (Supplementary Table 4). In region B, we found a cluster of nine overexpressed AltProts. Five are transcribed from ncRNA, two are located at the 5'UTR of mRNA, one at the 3'UTR and one results from a frameshift in the CDS. In region C, 45 AltProts are overexpressed: 24 from ncRNA, six from the 5'UTR, 10 from the 3'UTR, and five results from the frameshift in the CDS (Supplementary Table 4). Taken together, we identified several AltProts issued from ncRNA (~57%) which is in line with our previous work on a glioma cell line (NCH82)[33,34].

### Correlation between TCGA and proteomic data

We then compared our almost 5000 identified proteins to the TCGA database, on which 682 genes show an elevated expression in glioma; 282 proteins from these 682 genes were found in our samples (Supplementary Data 4). Of these 682 genes, 268 genes are suggested as prognostic indicators based on transcriptomic data from 153 patients; 201 genes are associated with an unfavorable prognosis, and 67 genes are associated with a favorable prognosis. In our proteomic data, we found 12 proteins associated with an unfavorable prognosis: 7 proteins are overexpressed in region A (CEND1, DMTN, PAK1, MAP2K1, THY1, VSNL1, and FN3KRP), 2 proteins are overexpressed in region B (AEBP1 and PDIA4), and 3 proteins are overexpressed in region C (POR, ERLIN2, and DBNL) (Table 2). We also found nine proteins associated with a favorable prognosis: seven proteins are overexpressed in region A (GLUD1, GDI2, SARS, SEPT2, PHGDH, KPNA3, and ARHGEF7), and two proteins are overexpressed in region C (PABPC1 and RBBP4) (Table 3).

### Integrating proteomics and survival data

Overall survival was associated with MGMT status (Supplementary Fig. 4a, c) and KPS (Supplementary Fig. 4a, b) but not with the extent of resection (Supplementary Fig. 4a, d). To find potential prognostic proteins from our proteomic data, we performed an ANOVA test on the entire proteomic dataset ($n = 46$ patients) according to OS. The cohort was divided into three groups according to the patient OS leading to 11 patients (25%) with OS > to the third quartile, 23 patients (50%) with an OS between the first and the third quartiles and 12 patients (25%) with an OS < to the first quartile were included in this analysis. 114 RefProts and 10 AltProts showed significance between these three groups of patients defined by their OS (Supplementary Data 5). Then, using a Cox model, 53 proteins (48 RefProts and 5 AltProts) were significant with a $P < 0.05$ (Supplementary Fig. 5), among which 28 proteins with a $P < 0.01$ (Table 4). After a step-by-step analysis and a bootstrap procedure, 5 proteins remained highly significantly correlated with survival: ALCAM, RPS14, ANXA11, PPP1R12A, and the AltProt IP_652563 (Fig. 3A). Based on the expression of these five proteins, two clusters of patients were identified (respectively cluster 1 and cluster 2) (Fig. 3B). The OS of the patients from the two clusters differed significantly (Fig. 3C and Supplementary Fig. 5). The expression of the 28 proteins with a $P < 0.01$ between patients of clusters 1 and 2 is shown Fig. 3D. 14 proteins are overexpressed in cluster 2 and associated with a poor prognosis (ANXA6, RPL11, HMGA1, IGHM, EIF3C, TUBA1A, GPHN, ANXA11, AP1G1, CDC42, PDCD6, IGHV3, IP_652563, and ALCAM). 14 proteins are overexpressed in cluster 1 and associated with a better prognosis (FXR1, RPS20, CALM3, S100B, CPNE6, RPS14, PPP1R12A, MTDH, WIBG, ACIN1, LASP1, THRAP3, PML, CDC5L). Among the five proteins highly correlated with survival based on the bootstrap procedure, IP_652563 is an AltProt issued from an ncRNA. This ncRNA is transcribed from the ENSG00000206028 gene which is expressed in glioma cell lines (Expression Atlas). This AltProt is

**Table 2 | Proteins associated with unfavorable prognostic in glioma and identified in regions A, B, and C**

| Uniprot | Gene description | Gene | Region |
|---|---|---|---|
| Q8N111 | Cell cycle exit and neuronal differentiation 1 | CEND1 | A |
| Q08495 | Dematin actin-binding protein | DMTN | A |
| Q13153 | P21 (RAC1) activated kinase 1 | PAK1 | A |
| Q02750 | Mitogen-activated protein kinase kinase 1 | MAP2K1 | A |
| P04216 | Thy-1 cell surface antigen | THY1 | A |
| Q9HA64 | Ketosamine-3-kinase | FN3KRP | A |
| P62760 | Visinin like 1 | VSNL1 | A |
| Q8IUX7 | AE-binding protein 1 | AEBP1 | B |
| P13667 | Protein disulfide isomerase family A member 4 | PDIA4 | B |
| P16435 | Cytochrome p450 oxidoreductase | POR | C |
| Q9UJU6 | Drebrin-like protein | DBNL | C |
| O94905 | Erlin-2 | ERLIN2 | C |

**Table 3 | Proteins associated with favorable prognostic in glioma and identified in regions A, B, and C**

| Uniprot | Gene description | Gene | Group |
|---|---|---|---|
| P00367 | Glutamate dehydrogenase 1 | GLUD1 | A |
| Q14155 | Rho guanine nucleotide exchange factor 7 | ARHGEF7 | A |
| P50395 | GDP dissociation inhibitor 2 | GDI2 | A |
| P49591 | Seryl-tRNA synthetase | SARS | A |
| Q15019 | Septin-2 | SEPT2 | A |
| O43175 | D-3-phosphoglycerate dehydrogenase | PHGDH | A |
| O00505 | Importin subunit alpha-4 | KPNA3 | A |
| P11940 | Poly(A) binding protein cytoplasmic 1 | PABPC1 | C |

a poor prognosis indicator whose expression is high in tumors of cluster 2. ALCAM and ANXA11 are the two other bad prognosis markers overexpressed in cluster 2. PPP1R12A and RPS14 are good prognosis markers overexpressed in cluster 1 (Fig. 3A). We confirmed the overexpression of the 48 markers in either cluster 1 or 2 based on the LFQ proteomic values. Figure 3E presents the quantification of 25 markers with a value <0.01. We further selected a panel of 10 markers (ANXA6, ANXA11, IGHM, RPS14, PPP1R12A, LASP1, ALCAM, CFH, HPSD1 and MAOB) for validation by immunohistochemistry on 23 representative tissues of the prospective cohort. For the AltProt, we could not perform this validation due to lack of antibodies for this type of protein. Representative images are presented in Fig. 4 and the results from fluorescence quantification in Fig. 5A. Interestingly, we confirm the overexpression of ANXA11, ANXA6, MAOB, IGHM and ALCAM in cluster 2 associated to poor prognosis and that PPP1R12A, PRS14, LASP1, HSPD1 and CFH in cluster 1 associated to better prognosis. Among the 10 markers only ALCAM expression variation is not found to be significative by immunofluorescence. In proteomics, the variation in the expression of ALCAM is also less significative than for the other markers ($P < 0.05$ only). Although not significant, a slight increase of fluorescence was observed in tumors of cluster 2 as well. The expression of ALCAM is associated with blood vessels as shown in Supplementary Fig. 6 and is known to participate in immune cell infiltration. Even though no difference in fluorescence was measured, blood vessels appeared to show different morphologies between patients of clusters 1 and 2 as shown in Supplementary Fig. 6. To confirm these findings, the five markers which were found to be the most significant from the proteomic experiments (Fig. 3A) were assessed by immunohistochemistry on an independent retrospective cohort of 50 patients (Fig. 5B). Patients were grouped according to their survival time, leading to three groups of different survival for this cohort. 13 patients

**Table 4 | Proteins associated with survival after Cox model $P < 0.01$**

| Parameter | Parameter estimate | Standard error | Chi-square | Pr > ChiSq | Hazard ratio | 95% Hazard ratio confidence limits | |
|---|---|---|---|---|---|---|---|
| IP_652563* | 0.28507 | 0.07527 | 14.3427 | 0.0002 | 1.33 | 1.147 | 1.541 |
| FXR1 | −1.25301 | 0.36448 | 11.8185 | 0.0006 | 0.286 | 0.14 | 0.584 |
| RPS20 | −0.7618 | 0.23552 | 10.4623 | 0.0012 | 0.467 | 0.294 | 0.741 |
| ANXA6 | 0.51404 | 0.15933 | 10.4094 | 0.0013 | 1.672 | 1.224 | 2.285 |
| ALCAM* | 0.56831 | 0.17706 | 10.3015 | 0.0013 | 1.765 | 1.248 | 2.498 |
| RPL11 | −0.78558 | 0.25342 | 9.6093 | 0.0019 | 0.456 | 0.277 | 0.749 |
| CALM3 | 0.25136 | 0.08171 | 9.4639 | 0.0021 | 1.286 | 1.096 | 1.509 |
| HMGA1 | −0.34607 | 0.11316 | 9.352 | 0.0022 | 0.707 | 0.567 | 0.883 |
| S100B | 0.24907 | 0.0818 | 9.2715 | 0.0023 | 1.283 | 1.093 | 1.506 |
| IGHM | 0.32975 | 0.10848 | 9.2399 | 0.0024 | 1.391 | 1.124 | 1.72 |
| EIF3C | −1.04772 | 0.34798 | 9.0653 | 0.0026 | 0.351 | 0.177 | 0.694 |
| CPNE6 | 0.33732 | 0.11439 | 8.6952 | 0.0032 | 1.401 | 1.12 | 1.753 |
| TUBA1A | 0.44037 | 0.15337 | 8.244 | 0.0041 | 1.553 | 1.15 | 2.098 |
| RPS14* | −0.64519 | 0.22592 | 8.1556 | 0.0043 | 0.525 | 0.337 | 0.817 |
| GPHN | 0.44548 | 0.15631 | 8.1221 | 0.0044 | 1.561 | 1.149 | 2.121 |
| ANXA11* | 0.27504 | 0.09713 | 8.0179 | 0.0046 | 1.317 | 1.088 | 1.593 |
| PPP1R12A* | −1.23054 | 0.43941 | 7.8424 | 0.0051 | 0.292 | 0.123 | 0.691 |
| AP1G1 | 0.83198 | 0.29958 | 7.7128 | 0.0055 | 2.298 | 1.277 | 4.134 |
| MTDH | −0.63924 | 0.2339 | 7.4688 | 0.0063 | 0.528 | 0.334 | 0.835 |
| WIBG | −0.58575 | 0.21444 | 7.4613 | 0.0063 | 0.557 | 0.366 | 0.848 |
| ACIN1 | −0.58334 | 0.21379 | 7.4451 | 0.0064 | 0.558 | 0.367 | 0.848 |
| LASP1 | −0.7371 | 0.27361 | 7.2578 | 0.0071 | 0.478 | 0.28 | 0.818 |
| THRAP3 | −0.49936 | 0.18628 | 7.1865 | 0.0073 | 0.607 | 0.421 | 0.874 |
| CDC42 | 0.51331 | 0.1933 | 7.0515 | 0.0079 | 1.671 | 1.144 | 2.44 |
| PDCD6 | 0.3683 | 0.13911 | 7.0097 | 0.0081 | 1.445 | 1.1 | 1.898 |
| PML | −0.37898 | 0.14353 | 6.9716 | 0.0083 | 0.685 | 0.517 | 0.907 |
| IGHV3_20 | 0.24137 | 0.09238 | 6.8269 | 0.009 | 1.273 | 1.062 | 1.526 |
| CDC5L | −0.52553 | 0.20365 | 6.6596 | 0.0099 | 0.591 | 0.397 | 0.881 |

*Proteins that remained significantly correlated to survival after step-by-step and bootstrap analyses.

had a low survival (less than 1 year), 25 patients had an intermediate survival (between 1 year and 2 years) and 12 patients had a high survival (more than 2 years). RPS14 and PPP1R12A were expressed at higher levels in the tumors of patients with longer survival compared to tumors of patients with low and intermediate survivals. ANXA11 was expressed at higher levels in tumors of patients with a low and intermediate survivals compared to patients with longer survival. No statistical differences of expression were observed for ALCAM expression, as observed for the first cohort of patients. These results confirm the validity of the identified prognostic markers, except for ALCAM.

## Discussion

In this work, we investigated the biology and heterogeneity of glioblastoma by a proteomic approach at a low spatial resolution to capture the tumor microenvironment. A non-targeted MALDI-MSI analysis followed by spatial segmentation using different algorithms allowed to highlight molecular heterogeneity among these tumors. We validated these observations with SpiderMass technology with a 93% good classification. Three sub-regions were identified (A- Red, B-Yellow, and C-Blue regions). To decode the biological pathways involved in these three regions, we performed a spatially resolved proteomic analysis that confirmed the data. Molecular signatures of different tumor subtypes were identified among the groups. From these data, we derived three molecular signatures. Region A is enriched in genes related to neurotransmission and synaptogenesis. Proteins overexpressed in region B are associated with immune infiltration while in

region C, we mainly identified proteins involved in RNA processing and metabolism.

Region A is associated with neuro-developmental genes, characteristic of neuronal/glial lineages or neural progenitor cells (NPC) (Fig. 2B). These included nervous system development markers (like CRMP family, GAP43, MAPT), oligodendrocyte development and differentiation markers (like ABI1, ASPA, CNP, CNTNAP1), stem and progenitor cell signatures (like TRIM2). The NPC-like state is correlated with markers for immature neurons (beta-3-tubulin), markers for mature neurons (NeuN) and markers indicative for synapses (synaptophysin, SV2A)[35]. In our data, we found Stathmin 1, NEFH, NEFM, and NEFL[36] which are also markers of the NPC-like state of the GSC. Region B is enriched in proteins linked to immune status with macrophages infiltration (Fig. 2C), such as complement factors, immunoglobulin heavy and light chains (IGHG2; IGKC; IGHM; IGHG1; IGLC6 and IGHA1), macrophage markers (CAPG) and coagulation cascade proteins (HP, KNG1, HRG, TTR, GCA, S100A9, STOM). In a study of Cheng et al.[37], eight immune-related genes (FOXO3, IL6, IL10, ZBTB16, CCL18, AIMP1, FCGR2B, and MMP9) were identified and used as unfavorable prognostic markers in glioblastoma. High-risk patients exhibited an enhanced intensity of local immune response compared to low-risk ones. From the 8 signature genes, AIMP2 was identified in region B, too. GSC markers but with a "stem-to-invasion" path were also identified in region B. CD44, NES and VIM, enriched in region B, are markers of the mesenchymal-like state.

The presence of class I self-antigen HLA proteins (HLA-A3 and HLA-B07) in group B is interesting since a positive correlation

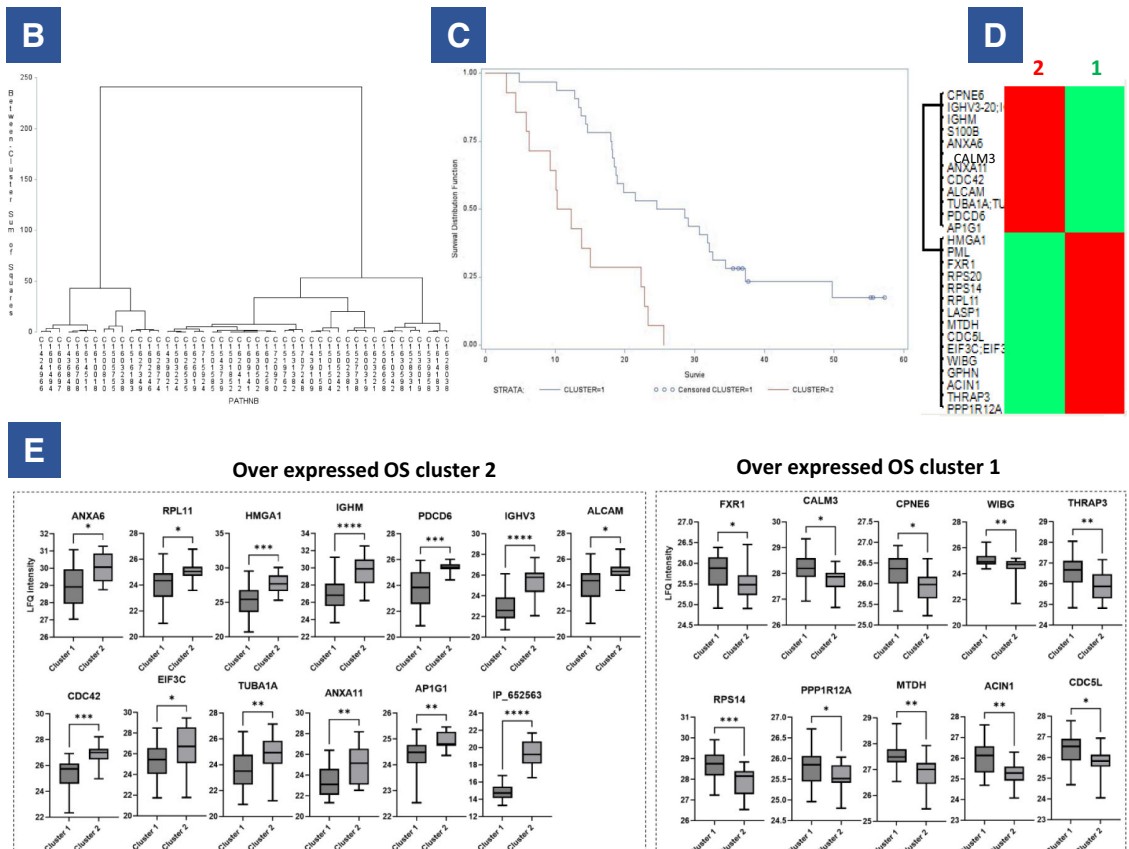

**Fig. 3 | Proteomic and survival analysis. A, B** Analysis of maximum likelihood estimates of the five proteins significantly correlated with survival (ANXA11, RPS14, ALCAM, PPP1R12A, and AltProt IP_652563) identified after a step-by-step analysis and bootstrap procedure and B. patient clustering based on these proteins. **C** Overall survival of the 46 patients according to the expression of the five prognostic markers. Two clusters of patients were identified with a clear difference in their survival. Cluster 1 has longer survival than cluster 2. **D** Heatmap of the 28 proteins significant in the Cox model ($P < 0.01$) between the two groups of patients defined by their OS (left). **E** Boxplots of the 28 prognosis proteins significant after applying the Cox model. Their LFQ values were compared between patients of cluster 1 (long survival, $n = 14$ patients) and cluster 2 (short survival, $n = 32$ patients). Significant differences were identified using two-sided unpaired $t$ test with ****$P < 0.0001$; ***$P < 0.001$; **$P < 0.01$, and *$P < 0.05$. Box plot indicates median, and whiskers indicate the extrema (minima and maxima values). The box extends from the 25th to the 75th percentiles. Exact $P$ value ANXA6 0.04; RPL11 0.0232; HMGA1 0.0007; IGHM < 0.0001; PDCD6 0.0001; IGHV3 < 0.0001; ALCAM 0.0232; CDC42 0.0002; EIF3C 0.0224; TUBA1A 0.0092; ANXA11 0.0016; AP1G1 0.0071; IP_ < 0.0001; FXR1 0.0234; CALM3 0.0149; CPNE6 0.0156; WIBG 0.0035; THRAP3 0.0062; RPS14 0.0009; PPP1R12A 0.0497; MTDH 0.0016; ACIN1 0.0034; CDC5L 0.0164. Source data are provided as a Supplementary Data file.

between HLA expression and some cancers has been demonstrated, such as cervical or nasopharyngeal carcinomas[38]. In a previous study based on HLA antigen frequencies in patients with glioma, patients positive for HLA-A*25 had a 3.0-fold increased risk of glioma ($P = 0.04$) and patients positive for HLA-B*27, a 2.7-fold risk ($P = 0.03$), compared with the control population. In contrast, the relationship between HLA-B*07 expression and higher risk to develop a glioma is very rare[39], as well as for HLA-A*3[40]. Taken together, these data confirmed that there is interpatient molecular heterogeneity that may be related to tumor phenotype and cellular plasticity[36] but not directly with transcriptional classification of

glioblastoma (proneural, neural, classical, and mesenchymal)[15]. Finally, systemic biology analyses revealed that group C is linked to an antiviral immune response and viral infection, in addition to RNA processing. Recent studies have reported a link between glioblastoma and perinatal viral exposure[41–44]. Further Epstein−Barr virus has been implicated in glioblastoma etiology[45]. Moreover, some studies have also reported that cytomegalovirus (CMV) promotes murine glioblastoma growth via pericyte recruitment and angiogenesis. In human, CMV nucleic acids and proteins have been observed within glioblastoma tumor tissue[46], although the link between glioblastoma and CMV remains very controversial[47].

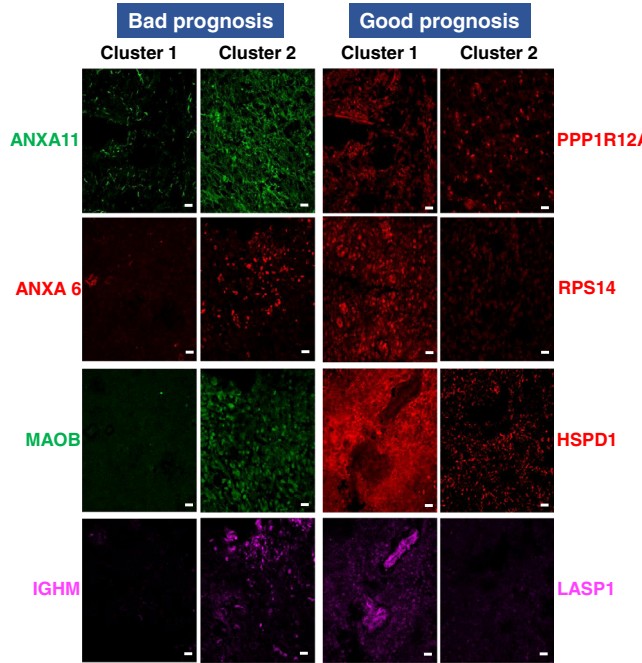

**Fig. 4 | Validation immunohistochemistry of the panel of survival markers identified.** Representative fluorescence images of eight proteins in the two OS clusters of patients. Images were acquired with a confocal microscope at ×40 magnification. The experiment was repeated on representative tissues of 23 patients for the prospective cohort and on representative tissues of 50 patients for the validation cohort. For each tissue, 3–4 images were taken for quantification. Scale bar = 20 μm.

Interestingly, a recent study has described the differential proteome and the molecular signatures associated to each histomorphologic niches of glioblastoma: cellular tumor, necrosis, infiltrating tumor (IT) and microvascular proliferations[48]. The IT region was associated with neuronal systems as well as stem cell-related pathways while the necrosis and MVP regions were associated with immune responses. A correlation can therefore be made with our study. Indeed, the region A we identified is linked to neurogenesis while the region B is linked to an intense immune response, and this was confirmed by several proteins found in common between IT vs region A (red) as well as between MVP/necrosis vs region B (yellow) (Supplementary Fig. 7A). However, contrary to the study of Lam et al., we have deliberately not determined our regions based on the histological annotations. As we have shown in this study, MALDI-MSI analysis have highlighted discrepancies between histological and molecular regions. Indeed, some regions identified as MVP-enriched, or necrosis were not molecularly different to the true tumor regions and inversely some tumor regions could be subdivided into several molecular areas. The three molecular regions we identified were histologically heterogeneous and could not be assigned to unique histological area. What is interesting is the fact that the tumor region defined by the pathologist could be subdivided into several molecular areas with defined molecular signatures. Some of these signatures can be enriched in some histological regions as demonstrated by Lam et al.[48], but can also be found in other histological niches as we have demonstrated here. To reinforce this argument, we have removed the micro-extracted points in the true necrotic area from our proteomic analysis and found the same clustering and the same molecular pathways involved (Supplementary Fig. 7B). Glioblastoma heterogeneity is therefore in part due to different histological microenvironments but also to the intrinsic heterogeneity in each specific microenvironment which make this disease even more complex.

The comparison with the TCGA specific glioma gene signature showed that 21 of them were associated with survival among the three groups identified in our study. Most of the proteins were identified in group A and are related to nervous system development, neuron differentiation, axon guidance. Three proteins were identified in group B and are linked to cytokine secretion, and five both in groups A & C related to Notch signaling. Notch signaling is an evolutionarily pathway that regulated important biological processes such as cell proliferation, apoptosis, migration, self-renewal, and differentiation. Growing evidence reveals that Notch signaling is highly active in glioma stem cells, in which it suppresses differentiation and maintains stem-like properties, contributing to glioblastoma tumorigenesis and conventional treatment resistance[49].

Taken together, we have revealed three main molecular regions in glioblastomas. Each region has a distinct molecular pattern, reflecting a specific molecular phenotype of the tumors. These different groups may be explained by an early differentiation due to the presence, in primary tumors, of subpopulation of cells with a distinct functional profile as well as the existence of cells with a high invasive potency. A recent study[50] proposed that glioblastoma stem cells (GSCs) acquire a high invasive activity through a mechanism called the 'stem-to-invasion path' and that long noncoding RNAs are one of the key factors. It has been demonstrated that these noncoding genomic regions can result in the synthesis of proteins, so-called alternative proteins, forming an unexplored ghost proteome with unknown function in cancer[51]. 170 alternative proteins (AltProts) were found significantly variable in the three groups identified above. Although the function of these AltProts remains poorly understood, they can have a role in the regulation of transcription and can also be present in extracellular vesicles[52]. Finally, more than 50% of the AltProts identified in the present study come from the translation of ncRNAs transcribed from pseudogenes. Seven AltProts have been identified in common with our previous study on the NCH82 glioma cell line, (IP_2323408 and IP_261897 described as an ncRNA and IP_755940, IP_593099, IP_774693, IP_572422 and IP_671464 from noncoding regions of mRNA. These last five AtlProts are pseudogenes for: HNRNPA1P30, TUBB2BP1, TUBAP2, TUBBP1, and TPI1P1, respectively. These pseudogenes, for which no protein has been observed yet, express their transcripts in glioma cell lines (Expression Atlas). Interestingly, the last one IP_079312, from the mRNA encoding EDARADD was correlated with a low survival rate in ovarian cancer patients[53]. Recently it has been demonstrated that pseudogenes can also be used as signatures for glioma prognosis. Six pseudogenes (SP3P, ANXA2P3, PTTG3P, LPAL2, CLCA3P, and TDH) were reported to be associated with overall survival in glioma. Nine other pseudogenes (TP73-AS1, AC078883.3, RP11-944L7.4, HAR1B, RP4-635E18.7, HOTAIR, SAPCD1-AS1, AC104653.1, and RP5-1172N10.2.) constitute a set of prognosis markers to predict survival of patients with glioma[54]. All these results provide insights into the biological role of pseudogenes in cancer and especially in glioma. Additionally, the identified AltProts translated from ncRNAs add additional information to the already known pseudogenes in glioma.

In another study, in which we studied interaction partners of AltProts in NCH82 cells[33], we identified five significantly different AltProts. One of them has been identified as overexpressed in region B: IP_156671 which originates from the 3'UTR of the transcript coding for SLC13A1. The four others are overexpressed in group C: IP_261897 coming from an ncRNA, IP_063564, IP_256988 both issued from the 3'UTR region of the CLDN19 and TBX21 genes, respectively, and IP_073718 originating from a shift in the reading frame of the CCDC181 gene. However, if the AltProts are indisputably detected thanks to mass spectrometry, the conventional techniques of intracellular detection allowing their follow-up (for example, with antibodies) are not realizable on a large-scale dimension. Thus case-by-case studies of the different AltProts identified are possible with the use of biochemistry allowing a combined expression of the AltProt of interest

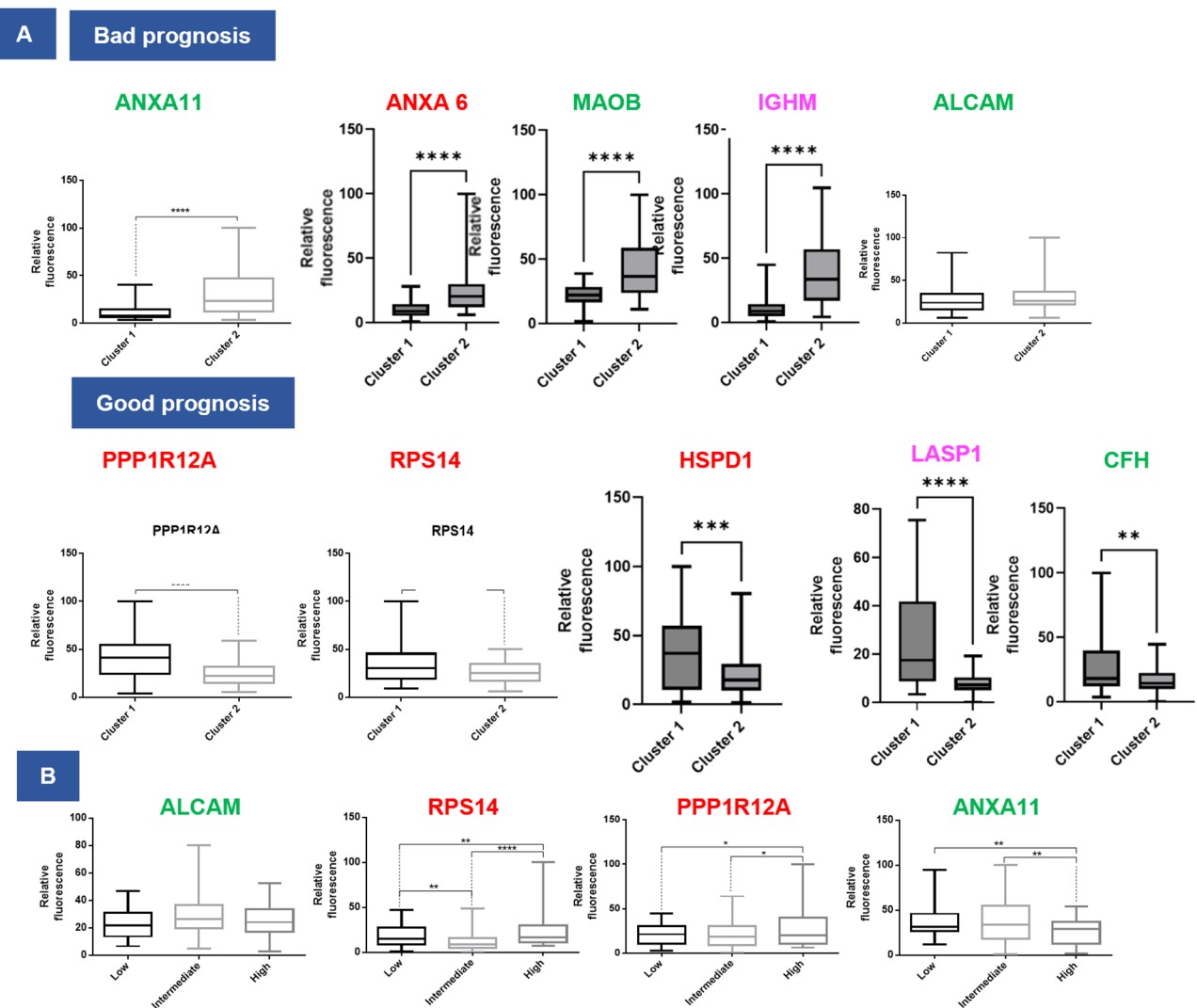

**Fig. 5 | Quantification of the panel of survival markers identified.** Quantification of fluorescence intensities of (**A**) the ten proteins in the two OS clusters. Images taken from 14 tumors of cluster 1 and 9 tumors of cluster 2 and (**B**) 4 proteins in an external cohort of glioblastoma patients (50 patients). Patients were classified according to their survival times (low, intermediate, and high). The fluorescence intensities of images taken from 50 tumors were quantified. For each tumor, 3–4 images were acquired and quantified. Significant differences were identified using multiple ANOVA comparison with ****$P$ < 0.0001; ***$P$ < 0.001; **$P$ < 0.01, and *$P$ < 0.05. Box plot indicates median, and whiskers indicate extrema (minima and maxima values). The box extends from the 25th to 75th percentiles.

with a tag (FLAG, HA…) for their monitoring. Similarly, the synthesis of an antibody directed against the AltProt of interest could be a major advance in the field.

Finally, the present study allowed us to identify prognostic proteins for glioblastoma: PPP1R12A and RPS14 are favorable prognostic markers while ALCAM, ANXA11, and AltProt IP_652563 are unfavorable prognostic markers. The expression of these markers was validated on an external independent cohort of 50 patients. These proteins were already identified as prognostic markers in lung and renal cancers (Human Protein Atlas). The expression of these markers in the tumors of patients with a low or intermediate survival (less than 2 years) was quite similar but differ from long survivors (from 2 to 5 years). We could therefore stratify them and give to the oncologist an indication of rapidity of the progression of the disease for each individual patient and guide the therapeutic decision. Indeed, some of the targets we have identified could be "druggable" such as ALCAM which is an unfavorable prognostic marker and which has also been found higher expressed in glioblastoma in another study[55]. Antibody–drug conjugates have been developed against ALCAM which could benefit for patients with a low survival[56].

In conclusion, we present here a spatial proteomic characterization in clinical samples of glioblastoma. The proteomic signatures we identified demonstrate the intratumoral molecular heterogeneity of glioblastoma tumors. While in previous studies, these signatures have been shown to be associated with survival[20], we showed that several of these signatures can be detected in a single tumor preventing their use as prognostic indicators. Despite this high heterogeneity, we have shown that some common markers could be identified for tumors of patients with inferior survival and inversely for tumors of patients with a longer survival, with validation on an external cohort of patients.

## Methods

Our research complies with all relevant ethical regulations. Approval of the study protocol was obtained from the Lille Hospital research ethics committee (ID-RCB 2014-A00185-42) before the initiation of the study. The study adhered to the principles of the Declaration of Helsinki and the Guidelines for Good Clinical Practice and is registered at NCT02473484. Informed consent was obtained from patients. Participants did not receive any compensation.

## Patient samples, consent, and ethics approval

Tumors from 96 patients were included in the study. 46 patients with newly diagnosed glioblastoma were prospectively enrolled between September 2014 and November 2018 at Lille University Hospital, France (NCT02473484). All patients gave written informed consent before enrollment. These 46 tumors were used for the proteomics analysis. Patients were adult, had no medical history of other cancers or previous cancer treatment, no known genetic disease potentially leading to cancer and no neurodegenerative disease. No other covariate characteristics were considered. Tumors samples were processed within 2 h after sample extraction in the surgery room to limit the risk of degradation of proteins. For the validation cohort used for IF analysis, 50 formalin-fixed paraffin-embedded (FFPE) glioblastoma tissues were obtained from the Pathology department of Lille Hospital, France. IDH-mutant tumors were excluded from the study.

## Deoxyribonucleic acid (DNA) extraction and quantification

Molecular analyses were performed on DNA-extracted FFPE tissues. The following tests were performed: Comparative genomic hybridization (CGH) array and assessment of $O^6$-methylguanine-DNA methyltransferase (MGMT) promoter methylation status. All tissues used for DNA extraction were histologically evaluated to determine the tumor cell content. Analyses were performed on all tissue samples. Samples with less than 40% of tumor cells content were considered as not interpretable when no molecular abnormalities were found. DNA extraction from FFPE was performed using the kit QIAamp DNA FFPE Tissue (Qiagen). CGH profiles were determined using a SurePrint G3 Human CGH Microarray Kit, 8x60K (Aligent) and the CytoGenomics v2.7 software. The limit of resolution was 1 Mb. The presence of 1p/19q codeletion, gain of chromosome 7, loss of chromosome 10, EGFR amplification and homozygous deletion of the cyclin-dependent kinase inhibitor 2 A (CDKN2A) gene was systematically evaluated. The MGMT promoter methylation status (CpGs 74-78) was determined after bisulfite treatment by pyrosequencing on a PyroMark Q96 with kit MGMT PyroMark (Qiagen). The presence of a methylation was score positive when a minimum of 8% of methylation was observed.

## MALDI mass spectrometry imaging (MALDI-MSI).

The 46 prospective tumors were analyzed by Matrix-Assisted Laser Desorption/Ionization (MALDI) mass spectrometry imaging. A Leica CM1510S cryostat (Leica Microsystems, Nanterre, France) was used to cut 12-µm sections in order to perform the MALDI-MSI analysis[57–60]. These tissue sections were deposited on ITO-coated glass slides (LaserBio Labs, Valbonne, France) and vacuum-dried during 15 min. Tissue sections were then soaked in different solutions to remove abundant lipids: (1) 1 min in 70% ethanol, (2) 1 min in 100% ethanol, (3) 1 min in acetone and (4) 30 s in chloroform with concomitant drying between washings. An electrospray nebulizer connected to a syringe pump (flow rate 180 nL/min) was used to uniformly spray a trypsin solution (60 µg/mL in NH4HCO3 50 mM) on the tissue surface for 15 min. ImagePrep (Bruker Daltonics, Bremen, Germany) was used as an incubation chamber by microspraying water heated to 37 °C for 2 h (60 cycles with 2 s spraying, 180 s incubation and 60 s drying using the nitrogen flow). For optimal digestion, a constant humidity atmosphere was maintained inside the spray chamber by filling a small container with 95 °C water. After digestion, HCCA/ANI[59] a solid ionic matrix was deposited using ImagePrep. Briefly, 36 µL of aniline were added to 5 mL of a solution of 10 mg/mL HCCA dissolved in ACN/0.1% TFA aqueous (7:3, v/v). A real-time control of the deposition is performed by monitoring scattered light to obtain a uniform layer of matrix. MALDI-MSI experiments were done on an Ultraflex II MALDI-TOF/TOF instrument (Bruker) with a smartbeam II solid-state laser using FlexImaging (version 2.1, Bruker Daltonics). Mass spectra were acquired using FlexControl (version 3.3.108, Bruker Daltonics) in positive reflector mode between 800 and 4000 $m/z$ range.

Recorded spectra were averaged from 400 laser shots per pixel acquired at 200 Hz laser repletion rate and with a 70 µm spatial resolution raster.

## MALDI-MSI data processing and analysis.

The MALDI-MSI data were analyzed using SCiLS Lab software (SCiLS Lab 2019, SCiLS GmbH). Common processing methods for MALDI-MSI were applied with a baseline removal using a convolution method and data were normalized using Total Ion Count (TIC) method[61,62]. Then, the resulting pre-processing data were clustered to obtain a spatial segmentation using the bisecting k means algorithm[63]. Different spatial segmentations were performed. First, an individual segmentation was applied to each tissue separately. Then, the data from all tissues were clustered together to obtain a global segmentation. Briefly, the spatial segmentation consists of grouping all spectra according to their similarity using a clustering algorithm and all pixels of a same cluster are color-coded. To limit the pixel-to-pixel variability, edge-preserving image denoising was applied. Note that a color is arbitrary assigned to a cluster and that several disconnected regions can have the same color, i.e. the same molecular content. The results of segmentation are represented on a dendrogram resulting from a hierarchical clustering. The branches of the dendrogram were defined based on a distance calculation between each cluster. The selection of different branches of the dendrogram will give a segmentation map where regions of distinct molecular composition were differentially color-coded. Individual segmentation provides information concerning the heterogeneity of the tissue section and the global segmentation is used to group patients with a similar molecular signature. For comparison, global segmentation was also performed using the Ward clustering method with IMAGEREVEAL MS Ver.1.1 (Shimadzu). The global spatial segmentation allowed to determine regions of interest (ROIs) which were then subjected to on-tissue microdigestion followed by microextraction for protein identifications.

## SpiderMass analyses.

The global design of the instrument setup has been described elsewhere[22]. Briefly, the system is composed of three parts including a laser system for micro-sampling of tissues set remotely, a transfer line allowing for transfer of the micro-sampled material to the third part, which is the mass spectrometer itself[64]. The first part is composed of a tunable wavelength OPO which is tunable from 2.8 to 3.1 µm (Radiant version 1.0.1, OPOTEK Inc., Carlsbad, CA, USA) pumped by a pulsed Nd:YAG laser (pulse duration: 5 ns, $\lambda = 1064$ nm, Quantel, Les Ulis, France). A biocompatible laser fiber (450 µm inner diameter; length of 1 m; Infrared Fiber Systems, Silver Spring, CO, USA) is connected to the laser system output and a handpiece including a 4 cm focusing lens is attached to the end of the laser fiber. The handpiece with a 4 cm focusing lens allows the user to hold the system and screen the surface of raw tissues at a resolution of 400 µm. In these experiments the irradiation time was fixed to 10 s at 4 mJ/pulse laser energy corresponding to a laser fluence of ~3 J/cm². The laser energy was measured at the focal point of the focusing lens using a power meter (ThorLabs, Maisons-Laffitte, France). The second part of the system corresponds to a 2-m length transfer line made from a Tygon ND 100-65 tubing (2.4 mm inner diameter, 4 mm outer diameter, Akron, Ohio, USA). The transfer line is attached on one side onto the laser handpiece at the end of the laser fiber and on its other side directly connected to the mass spectrometer (Xevo, Waters, Manchester, United Kingdom) from which the conventional electrospray source was removed and replaced by an atmospheric pressure interface[64]. Each acquisition was accompanied by a 150 µL/min isopropanol infusion. Spectral acquisition was performed both in positive and negative ion resolution mode with a scan time of 1 s. Prior to SpiderMass analysis, the samples were taken out of the −20 °C freezer and thawed to RT for 30 s. The spectral acquisition sequence was composed of two or

three acquisitions using 1-s irradiation periods. The ROI were selected using the morphological controls and acquired peptide MALDI-MSI data prior to each SpiderMass acquisition to ensure that each acquisition was performed on the same histological area[23].

## Classification model construction

For data analysis, all raw data files produced with the SpiderMass instrument were imported into the Abstract Model Builder (AMX v.0.9.2092.0) software. After importation, spectra were first pre-processed. The pre-processing steps include background subtraction, total ion count normalization, lockmass correction and re-binning to a 0.1 or 0.2 Da window. All processed MS spectra obtained from the 30 histologically validated samples were then used to build a principal component analysis and linear discriminant analysis (PCA-LDA) classification model[23]. The first step consisted of PCA to reduce data multidimensionality by generating features that explain most of the variance observed. These features were then subjected to supervised analysis using LDA by setting the classes that the model will be based upon. LDA attempts to classify the sample spectra and assess the model by cross-validation. Cross-validation was carried out by either using the "20% out" or the "leave-one patient-out" methods. For the first method, 20% of MS spectra are randomly taken from the total spectra and the model is reconstructed from the remaining 80%. The remaining 20% of spectra are used to interrogate the reconstructed model. The permutation is automatically reiterated for five cycles before reporting the cross-validation results. For the second method, the spectra are grouped by patient and left out one by one; at each step the model without the patient is interrogated against this model.

**Spatially resolved proteomics.** *On-tissue digestion*: A total of 122 ROIs in the 46 tumors were selected from MALDI-MSI. Spatially resolved microproteomics was performed on the predefined ROIs according to the previously published protocol[65]. Briefly, tissue sections of 20 µm thickness were cut and subjected to different washes to remove lipids. Then, on-tissue digestion is performed using a LysC-trypsin solution (40 µg/mL in Tris-HCl 50 mM, pH 8.0). This solution was deposited using a piezoelectric microspotter (CHIP-1000, Shimadzu, CO, Kyoto, Japan) on each ROIs with a total area of 1 mm² (4 × 4 spots of 200 µm. Enzyme droplet was maintained for a total of 2 h digestion. After enzyme deposition 0.1% TFA was spotted for 25 cycles with 100 pL on each spot/cycle.

*Microextraction by liquid microjunction*: After tissue microdigestion, the tryptic peptides were extracted using an automated platform, the TriVersa Nanomate platform (Advion Biosciences Inc., Ithaca, NY, USA) with Liquid Extraction Surface Analysis (LESA) option[65]. Briefly, a volume of solvent was aspirated onto a tip and dispensed onto the digested region. The droplet formed was maintained between the tip and the tissue and then aspirated after 15 s. The recovery solution is finally pooled in a low-binding tube. Three extractions steps were performed per region using different solutions: (1) 0.1% TFA, (2) ACN/0.1% TFA (8:2, v/v), and (3) MeOH: 0.1% TFA (7:3, v/v). Two extraction cycles per point were performed to increase the amount of material collected.

*NanoLC-MS & MS/MS analysis*: Prior to MS analysis, the reconstituted samples were desalted using C18 Ziptip (Millipore, Saint-Quentin-en-Yvelines, France), eluted with 80% ACN and vacuum-dried. The dried samples were resuspended in 0.1% FA aqueous/ACN (98:2, v/v). Peptides separation was performed by reverse phase chromatography, using a NanoAcquity UPLC system (Waters) coupled to a Q-Exactive Orbitrap mass spectrometer (Thermo Scientific) via a nanoelectrospray source. A pre-concentration column (nanoAcquity Symmetry C18, 5 µm, 180 µm × 20 mm) and an analytical column (nanoAcquity BEH C18, 1.7 µm, 75 µm × 250 mm) were used. A 2 h linear gradient of acetonitrile in 0.1% formic acid (5%-35%) was applied, at the flow rate of 300 nl/min. For MS and MS/MS Acquisition (Xcalibur 4.1

and Exactive Series 2.9), a data-dependent mode was defined to analyze the 10 most intense ions of MS analysis (Top 10). The MS analysis was performed with an $m/z$ mass range between 300 and 1600, a resolution of 70,000 FWHM, an AGC of 3e6 ions and a maximum injection time of 120 ms. The MS/MS analysis was performed with an $m/z$ mass range between 200 and 2000, an AGC of 5e4 ions, a maximum injection time of 60 ms and the resolution was set at 17,500 FWHM. To avoid any batch effect during the analysis, the extractions were chosen at random to create analysis sequences.

*Data analysis*: All MS data were searched with MaxQuant software[66,67] (Version 1.5.3.30) using Andromeda search engine[68] against the complete proteome for *Homo sapiens* (UniProt, release July 2018, 20,412 entries). Trypsin was selected as enzyme and two missed cleavages were allowed, with N-terminal acetylation and methionine oxidation as variable modifications. The mass accuracies were set to 6 ppm and 20 ppm, respectively, for MS and MS/MS spectra. False discovery rate (FDR) at the peptide spectrum matches (PSM) and protein levels was estimated using a decoy version of the previously defined databases (reverse construction, Homo sapiens, UniProt, release July 2018) and set to 1%. A minimum of two peptides with at least one unique is necessary to complete the identification of a protein. The MaxLFQ algorithm[69] was used to performed label-free quantification of the proteins. The resulting file was analyzed using Perseus software (version 1.6.0.7). First, hits from the reverse database, proteins with only modified peptides and potential contaminants were removed. Statistical analyses were performed using ANOVA with a truncation value based on "Benjamini Hochberg FDR" of 5%. Three categorical annotation groups were used for the ANOVA, i.e. (1) the color group based on the three colors from Scils global segmentation of the 46 samples (Red; Yellow and Blue), (2) the patient groups which are determined by the main color present in each tumor sample (Groups A, B, C), and (3) the patients' survival time (patients with an OS > to the third quartile, patients with an OS between the first and the third quartile and patients with an OS < to the first quartile). Proteins significantly different were selected and normalized by a Z-score with matrix access by rows. For representation, a hierarchical clustering was performed using the Euclidean parameter for the distance calculation, and the average option for linkage in the rows and columns of the trees with a maximum of 300 clusters.

## System biology analyses

Annotation analysis of gene ontology terms for the identified proteins were performed using PANTHER Classification System (version 14.1, http://www.pantherdb.org), FunRich (Version 3.1.3)[70] and the STRING database (version 11.0, www.string-db.org)[71]. Potential interaction network was then loaded into Cytoscape 3.7.2 with relative expression data using Idmapper[72]. The Reactome FI plugging was used to select a subnetwork of gene ontology terms and NCI database-associated disease-specific proteins. The relationships between the differentially expressed proteins among all conditions were also depicted based on the Ariadne ResNet database using Elseviers' Pathway Studio (version 11.0, Elsevier). The subnetwork Enrichment Analysis (SNEA) algorithm was used to detect the statistically significant altered biological pathways in which the identified proteins are involved.

## Human pathology atlas

The glioma data contained in the Human pathology atlas[73] were used. Based on TCGA transcriptomics and antibody-based protein data from 153 patients, this database identified 268 potentially prognostic genes (201 unfavorable and 67 favorable prognoses). These data were compared to the proteins identified in our study.

## Alternative protein identification

RAW data obtained by nanoLC-MS/MS analysis were analyzed using Proteome Discoverer V2.3 (Thermo Scientific) LFQ quantification with

the following parameters: trypsin as enzyme, two missed cleavages, methionine oxidation as variable modification and carbamidomethylation of cysteines as static modification, Precursor Mass Tolerance: 10 ppm and fragment mass tolerance: 0.6 Da. The validation was performed using Percolator with a FDR set to 0.001%. A consensus workflow was then applied for the statistical arrangement, using the high-confidence protein identification. The protein database was uploaded from Openprot (https://openprot.org/) and included RefProt, isoforms and AltProts predicted from both Ensembl and RefSeq annotations (GRCh38.83, GRCh38.p7)[32,74,75] for a total of 658263 entries. The identified abundance was extracted to PD2.3 and loaded in Perseus to perform statistical analysis and graphical representation.

## Statistical analyses

Descriptive analyses were performed on clinical data. Patients were divided into three groups according to the quartiles of overall survival (<Q1, Q1–Q3, > Q3). The Cox model was used to determine which proteins were most associated with overall survival. Stepwise analysis and bootstrap methods (500 samples) were used to guarantee the robustness of the results. The proteins selected after this step were used to carry out a hierarchical classification (Euclidean distance and Ward's method) on the 46 patients to determine if there were any subgroups (clusters). Finally, the clinical variables were analyzed according to the different clusters to provide a clinical description of the clusters obtained. Statistical analyses were performed using the SAS Software, V9.4.

## Confirmatory immunohistochemistry analyses

Survival group validation was performed using antibodies directed against ALCAM, RPS14, ANXA11, and PPP1R12A. The tissues were incubated with a primary antibody at 4 °C overnight, followed by the application of a secondary antibody (Alexa fluor conjugated antibody, donkey anti-rabbit 555, donkey anti-mouse 647, donkey anti-goat 488, donkey anti-mouse 488, 1/200 dilution) for 1 h at RT. For the validation cohort, dewaxing and antigen retrieval with citrate buffer were first performed before the incubation with the antibodies. We used the following primary antibodies: ALCAM (R&D Systems; 1/40 dilution), RPS14 (Invitrogen, 1/100 dilution), ANXA11 (OriGene, 1/100 dilution), PPP1R12A (Invitrogen, 1/250 dilution), MAOB (Abbexa, 1/100), ANXA6 (Abcam, 1/50), IGHM (Abcam, 1/50), HSPD1 (Abcam, 1/200), LASP1 (Santa Cruz Biotechnology, 1/50), CFH (Abcam, 1/200). All slides were imaged on the Zeiss LSM700 confocal microscope. Three to four pictures were taken for each tumor section. Processing of the images and fluorescence intensity quantification was performed using ImageJ software.

## Reporting summary

Further information on research design is available in the Nature Research Reporting Summary linked to this article.

## Data availability

The data generated in this study, including MS raw files, MaxQuant files, and annotated MS/MS datasets, have been deposited to the ProteomeXchange Consortium Via PRIDE partner repository with the accession code PXD016165 The public data used in this study were downloaded from The Cancer Genome Atlas Program (TCGA) and are accessible at the Data Coordinating Center (DCC) for public access [http://cancergenome.nih.gov/], and the Human Protein Atlas [https://www.proteinatlas.org/] Database containing Human proteins sequences is accessible at Uniprot [https://www.uniprot.org/] and at OpenProt [https://openprot.org/] for Alternative proteins. The remaining data are available within the Article, Supplementary Information, or Source Data file. Source data are provided with this paper.

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

## Acknowledgements

This research was supported by grants from the Ministère de L'Education Nationale, de L'Enseignement Supérieur et de la Recherche, ANR (I.F.), SIRIC ONCOLille (M.S.), Grant INCa-DGOS-Inserm 6041aa (I.F. and M.S.), and INSERM, Ligue Contre le Cancer (E.L.).

## Author contributions

Conceptualization: M.S., I.F., and E.L.R.; methodology: M.D., M.Wi., L.D., M.S., I.F., J.P., S.A., and N.O.; software: L.D., M.Wi., T.C., and F.Z.; validation: M.D., L.D., C.A.M., and F.E.; formal analysis: M.S., M.D., L.D., T.C., M.Wi., and P.D.; investigation: M.D., M.Wi., L.D., E.L.R., I.F., and M.S.; data curation: M.Wi., M.S., L.D., and M.D.; writing: M.S., M.D., E.L.R., M.We., T.C., L.D., and M.Wi; original draft: M.S., L.D., M.D., and M.Wi.; supervision and project: M.Wi., E.L.R., I.F., and M.S.; administration: E.L.R., M.S., and I.F.; funding acquisition: I.F., M.S., and E.L.R.

## Competing interests

Dr. E.L.R. has received honoraria for lectures or advisory board from Adastra, Bayer, Janssen, Leo Pharma, Pierre Fabre, and Seattle Genetics. Dr. M.W. has received research grants from Apogenix, Merck, Sharp & Dohme, Merck (EMD), Philogen and Quercis, and honoraria for lectures or advisory board participation or consulting from Adastra, Bayer, Bristol Meyer Squibb, Medac, Merck, Sharp & Dohme, Merck (EMD), Nerviano Medical Sciences, Novartis, Orbus, Philogen and y-Mabs. The remaining authors declare no competing interests.
