## [Peer Review File · Nature Communications]

Spatial analysis of the glioblastoma proteome reveals specific molecular signatures and markers of survivalReviewers' Comments:

Reviewer #1:

Remarks to the Author:

In the manuscript by Duhamel et al. entitled "Spatial reference and alternative proteome analysis of glioblastoma reveals molecular signatures and associates survival with specific markers", the authors perform a relevant study that enables the characterization of molecular regions in glioblastoma tissue based on protein expression.

Results are relevant and inside of the scope of the journal.

Authors globally characterize by maldi mass spectrometry imaging and spatial proteomics analysis glioblastoma. The authors report several interesting results that should deserve publication in Nat Communications, as 5 proteins as survival markers and the presence of 10 proteins associated to survival from AltORFs or non-coding RNA.

1. The abstract should be edited to indicate in detail the techniques used in the manuscript. The authors just indicate "spatially-resolved high resolution mass spectrometry proteomics." that barely indicates which techniques are used in the manuscript.

2. The reviewer acknowledges the difficulty of validating the ghost proteome. Indeed, cross-linking proteomics might help in their validation. Although out of the manuscript, the authors should acknowledge this problem in the discussion beyond there were no antibodies for validation, and discuss how they could be validated.

3. More importantly and also related to point 2, although the work support the conclusions and claims, the authors could use the CPTC data of glioblastoma (TMT 11-plex) for validation of the ghost proteome. Indeed, if there were enough data of clinical samples regarding survival, the authors could validate using another dataset the findings observed in the presented manuscript regarding their association to prognosis.

Reviewer #2:

Remarks to the Author:

This is an interesting paper in which the authors used MALDI-based spatial proteomic analysis on a cohort of 96 glioblastoma patients with survival varying from few months to >4 years. Of the entire cohort, 46 tumors were analyzed by spatially-resolved high resolution MS proteomics identifying three major molecular profiles associated with immune, neurogenesis and tumorigenesis signatures. Many of these regions co-occured within the same samples (intra-tumoral heterogeneity) and were not carefully histologically annotated. Follow-up experiments showed that some of the identified proteins correlated with patient's survival, which the authors also explore in an external cohort of 50 GBM samples. Unfortunately, despite the massive differences in survival of this cohort, the differences during the external validation were mild and didn't show a dose-dependant effect oftentimes.

Comments:

1) The spatially resolved nature of the proteomics provide a very complex and well-executed experiment, but one wonders if they have specific confounding histological correlates. The authors state there was poor overlap between histology annotations and MALDI selected regions, but it was unclear if this meant tumor pattern 1 vs pattern 2 or Tumor vs Necrosis vs normal brain. This is absolutely critical for the proper evaluation of the results.

2) Some of the "X"-labeled macro-extracted points in the MALDI labeled slides overlap with regions the pathologist seems to have labeled as necrosis or infiltrating tumor region. Can this be more explicitly discussed, as it seems vital to interpretation. One worries that the major partitions identified are those of regional histological patterns not directly related to intrinsic tumor biology (see point 3)

3) There are relevant studies of intra-tumor heterogeneity both at the RNA (PMID: 29748285) and

protein level (PMID: 35013227). These could be discussed and integrated with the results, especially since these profiles were correlated with histological patterns. The authors do indeed note that some of their identified biomarkers labeled blood vessels, which suggests that some of these regions are due to microenvironments differences and not true glioma biology.

4) Can you ensure the Group A is not regions that correspond to infiltrating brain tissue given the process of neurogenesis/synapse function? Group B also sounds very suspicious for necrotic tissue and areas of microvascular proliferation (PEC?).

5) The survival proteins chosen seem to show only slight differences. The external validation is seen as a positive for the study, but the overall differences are minor given the massive stratification of outcomes (months vs >4 years). It is hard to know what clinical value differences, even if slightly significant at the population level, means for individual patients.

Minor

1) Page 8, Line 193 - Guidedec → guide

2) The reference to 96 vs 46 + 50FFPE is confusing. The cohort for the proteomics component was 46 and the additional 50 were for validation/testing. This should be more clearly communicated. The methods later go on to say only 30 histologically validated samples were used. While the exact number likely doesn't change results, this is the true number used in the analysis and should be highlighted more.

3) PEC should be more explicitly defined in the manuscript

Reviewer #1 (Remarks to the Author): Expert in spatial proteomics

In the manuscript by Duhamel et al. entitled “Spatial reference and alternative proteome analysis of glioblastoma reveals molecular signatures and associates survival with specific markers”, the authors perform a relevant study that enables the characterization of molecular regions in glioblastoma tissue based on protein expression.

Results are relevant and inside of the scope of the journal. Authors globally characterize by maldi mass spectrometry imaging and spatial proteomics analysis glioblastoma. The authors report several interesting results that should deserve publication in Nat **Communications**, as 5 proteins as survival markers and the presence of 10 proteins associated to survival from AltORFs or non-coding RNA.

1. The abstract should be edited to indicate in detail the techniques used in the manuscript. The authors just indicate “spatially-resolved high resolution mass spectrometry proteomics.” that barely indicates which techniques are used in the manuscript.

First, we would like to thank the reviewers for their valuable comments.

We have added a sentence in the abstract to detail the mass spectrometry approaches we have used for the study.

2. The reviewer acknowledges the difficulty of validating the ghost proteome. Indeed, cross-linking proteomics might help in their validation. Although out of the manuscript, the authors should acknowledge this problem in the discussion beyond there were no antibodies for validation, and discuss how they could be validated.

We have added a paragraph in the discussion describing the limitations of current techniques and how case-by-case studies could be done lines 806-812 : " However, if the AltProts are indisputably detected thanks to mass spectrometry, the conventional techniques of intracellular detection allowing their follow-up (for example with antibodies) are not realizable on a large scale dimension. Thus case-by-case studies of the different AltProts identified are possible with the use of biochemistry allowing a combined expression of the AltProt of interest with a tag (FLAG, HA...) for their monitoring. Similarly, the synthesis of an antibody directed against the AltProt of interest could be a major advance in the field."

To respond to the reviewer’s proposal about the use of cross-linking proteomics for the validation of AltProts, PRISM laboratory is a pioneer in its use combined with the identification of AltProts and has published several times on the usefulness of the cross-linking method in the identification of AltProts’ partners to find the signaling pathways implicating them (PMID: 31128158, PMID: 32324228).. However, in the present study, several points limit the use of such a method:

- Here we have carried out proteins identification from liquid micro-extraction on tissues. So even if numerous advances in cross-linking technique exists today to reduce the

quantity of material used, no approach has yet been developed to deal with such small quantities of material directly on tissue sections.

- Finally in the present study, we have identified precise targets. There is no guarantee that the cross-linking strategy will allow us to find these targets precisely, since the cross-linking technique is the only non-targeted interactomics strategy. It would be therefore more appropriate to use targeted biochemistry methods such as the BioID or pull-down after expression with a TAG for example. Indeed, we have already performed TurboID to identify the partners of the AltProt correlated to the survival of glioblastoma patients. These results were not included in the manuscript because we plan to include them in another manuscript in which we will focus on the function of this AltProt in glioblastoma. You can find in the table below the most potent partners we have identified. These proteins are mainly involved in translation events, ribosomal assembly and Wnt pathway.

Protein names	Log2 fc
Nucleolar and spindle-associated protein 1	3.2
Nucleolar protein 7	2.3
Cysteine-rich PDZ-binding protein	2.5
Eukaryotic translation initiation factor 1	2
E3 ubiquitin-protein ligase Midline-1	2.6
Notchless protein homolog 1	2.2
Zinc finger protein 706	2.0
Collagen alpha-1(XIV) chain	2.0
Protein phosphatase 1 regulatory subunit 21	1.9
NADH dehydrogenase [ubiquinone] flavoprotein 2, mitochondrial	1.7
Proline/serine-rich coiled-coil protein 1	1.7
Protein downstream neighbor of Son	2.0
Tripartite motif-containing protein 2	2.6
YrdC domain-containing protein, mitochondrial	2.0
PDZ and LIM domain protein 7	2.0
Coiled-coil and C2 domain-containing protein 1A	1.4
Thioredoxin domain-containing protein 5	2.0
Casein kinase I isoform delta	1.5
Casein kinase I isoform epsilon	1.4
Translation machinery-associated protein 16	1.6
40S ribosomal protein S28	1.3
ER membrane protein complex subunit 3	2.2
Trifunctional enzyme subunit alpha, mitochondrial	2.0
Carnitine O-acetyltransferase	2.0
28S ribosomal protein S11, mitochondrial	1.3
Signal recognition particle 14 kDa protein	1.2
40S ribosomal protein S30	1.1
Ribosome-recycling factor, mitochondrial	1.2
RNA-binding protein 42	2.2
Leucine-rich repeat-containing protein 47	1.3
40S ribosomal protein S19	1.2
Nucleolar protein 16	1.1
60S ribosomal protein L27a	1.2

39S ribosomal protein L55, mitochondrial	1.4
DDB1- and CUL4-associated factor 13	2.4
Pre-mRNA-splicing factor 38B	2.2
Cleavage stimulation factor subunit 1	1.7
ER membrane protein complex subunit 2	1.9
WD repeat-containing protein 6	1.7
NADH dehydrogenase [ubiquinone] 1 alpha subcomplex subunit 12	1.7
10 kDa heat shock protein, mitochondrial	1.6
Golgi reassembly-stacking protein 2	1.9
Zinc transporter 6	2.7
Zinc transporter 5	2.4
FAS-associated factor 2	1.6
Sphingomyelin phosphodiesterase 4	1.7
Nucleoside diphosphate-linked moiety X motif 19	3.0
Ubiquitin carboxyl-terminal hydrolase 30	2.7
Tudor and KH domain-containing protein	1.9
Serine/threonine-protein kinase VRK2	1.2
AFG3-like protein 2	1.3
Aldehyde dehydrogenase family 3 member A2	1.6
UBX domain-containing protein 4	1.7
OCIA domain-containing protein 1	2.0
ADP-ribosylation factor GTPase-activating protein 3	1.2
Golgi resident protein GCP60	1.6
Inner nuclear membrane protein Man1	1.5

3. More importantly and also related to point 2, although the work support the conclusions and claims, the authors could use the CPTC data of glioblastoma (TMT 11-plex) for validation of the ghost proteome. Indeed, if there were enough data of clinical samples regarding survival, the authors could validate using another dataset the findings observed in the presented manuscript regarding their association to prognosis.

We thanks the reviewer for his/her idea. The datasets mentioned in his/her comment are currently under embargo, and the RAW data are not yet usable for publication. However, we are currently in discussion with a bioinformatics team, in order to carry out a large-scale study on the detection of AltProts in public domain data, using up to date techniques, algorithms and software to identified AltProts specifically. We intend to target several cancers including glioblastoma.

Reviewer #2 (Remarks to the Author): Expert in bioinformatics, proteomics, and glioblastoma

This is an interesting paper in which the authors used MALDI-based spatial proteomic analysis on a cohort of 96 glioblastoma patients with survival varying from few months to >4 years. Of the entire cohort, 46 tumors were analyzed by spatially-resolved high resolution MS proteomics identifying three major molecular profiles associated with immune, neurogenesis and tumorigenesis signatures. Many of these regions co-occured within the same samples (intra-tumoral heterogeneity) and were not carefully histologically annotated. Follow-up experiments showed that some of the identified proteins correlated with patient's survival, which the authors also explore in an external cohort of 50 GBM samples. Unfortunately, despite the massive

differences in survival of this cohort, the differences during the external validation were mild and didn't show a dose-dependant effect oftentimes.

Comments:

1) The spatially resolved nature of the proteomics provide a very complex and well-executed experiment, but one wonders if they have specific confounding histological correlates. The authors state there was poor overlap between histology annotations and MALDI selected regions, but it was unclear if this meant tumor pattern 1 vs pattern 2 or Tumor vs Necrosis vs normal brain. This is absolutely critical for the proper evaluation of the results.

First, we would like to thank the reviewers for their valuable comments.

In the MALDI images, we keep the heterogeneity of the tissues. Therefore, in some cases, we can see a correlation between the molecular regions identified in MALDI and the histological regions annotated by the pathologist (cases 4 and 7 for example) but in other and in the majority of the cases, we can observe a poor overlap within the tumor regions annotated by the pathologist (cases 8, 10, 12, 18, 19, 23, 24, 28, 30, 31, 34, 38, 39, 44 ...). For some other cases, necrotic regions or regions with MVP identified by the pathologist were not detected as different as the tumor region by MALDI (cases 3, 14, 21, 26, 36, 45). We therefore add additional information with the MALDI molecular analysis to the histological annotations, which could help the pathologist. The micro-extracted points were selected within these MALDI regions without preconceptions about the histological regions. We have considered that the glioblastoma tumor microenvironment is inherently heterogeneous and all the histological features are part of the pathology and participate to glioblastoma progression and resistance in many aspects. That is why we choose to analyze them in our study. We have added these information in the results section (lines 475-483).

2) Some of the "X"-labeled macro-extracted points in the MALDI labeled slides overlap with regions the pathologist seems to have labeled as necrosis or infiltrating tumor region. Can this be more explicitly discussed, as it seems vital to interpretation. One worries that the major partitions identified are those of regional histological patterns not directly related to intrinsic tumor biology (see point 3)

With the pathologist, Claude Alain Maurage, we have redefined some areas on the tissues which were not fully annotated and we have also homogenized the annotations. At the end, we have observed 7 different regions: tumor, very dense infiltration, dense infiltration, tumor + necrosis, microvascular proliferation (MVP) + tumor, tumor + infiltration and necrosis. I have next calculated the percentage of micro-extracted points in each of these regions for the 3 groups we have identified by mass spectrometry (see table below).

Region	Tumor	very dense infiltration	dense infiltration	necrosis+tumor	MVP+tumor	tumor+inflammation	necrosis
yellow	45	0	0	25	5	0	20
red	39.3	14.3	28.6	10.7	3.6	3.6	0
blue	73.6	5.7	2.3	3.4	12.6	1.1	1.1

% of micro-extracted points in the histologic regions annotated by the pathologist for each group (red, blue, yellow).

As you can see, the majority of the micro-extracted points are in the tumor tissue for the three

groups. The blue and the red regions are more heterogeneous, comprising micro-extracted points in all histological regions. The yellow group is more homogeneous with points located in the tumor, in areas of tumor with few necrotic cells and in necrotic regions. This table has been added as supplementary Table 1 and some additional explanations have been added in the results section (lines 523-532).

Moreover, to verify that the histological regions do not influence the proteomics clustering. We have removed the micro-extracted points in the purely necrotic areas from the proteomics analysis. 4 of these points belong to the yellow group, while 1 belongs to the blue group. The grouping is highly similar to the one we have presented in the manuscript. 180 proteins are overexpressed in the yellow group and when comparing the biological pathways, we can see a high similarity between the yellow group with or without the points micro-extracted in the necrotic part.

3) There are relevant studies of intra-tumor heterogeneity both at the RNA (PMID: 29748285) and protein level (PMID: 35013227). These could be discussed and integrated with the results, especially since these profiles were correlated with histological patterns. The authors do indeed note that some of their identified biomarkers labeled blood vessels, which suggests that some of these regions are due to microenvironments differences and not true glioma biology.

These two studies are very interesting and informative and as you can see on the Venn Diagrams below, we found correlations between the PMID: 35013227 and our proteomics analyses. The red region which is characterized by neurogenesis share 81 proteins with the infiltrating tumor area of the study of Lam *et al.* (around 50 % of the proteins of the study of Lam). The yellow

region characterized by immune response enrichment share 30 proteins with the MVP and necrosis areas of the study of Lam *et al.* (around 30% of the proteins of the study of Lam). But as you can see in the table above, each region we identified in the present study is not defined by only one histological area. They are composed of points micro-extracted in several histological areas. Therefore, we cannot make a direct link with the histology. Nevertheless, we have included these results in the revised version of the manuscript and we have discussed them as well (lines 737-760).

What is also interesting to notice is that in one given histological region, we can also observe a certain heterogeneity. For example, in the tumor part including necrotic cells, 5 out of 11 points are in the yellow group and 6 out of 11 points are in the red (x3) and blue groups (x3). Heterogeneity is not only associated to the true tumor area. We have discussed this point in the revised manuscript.

We believe that glioma biology is not independent of the microenvironment. Many studies have shown that the microenvironment takes an integral part in tumor development. From a histological point of view, glioblastoma features include true tumor, infiltrating tumor, microvascular proliferation and necrosis. Many studies have focused on the tumor cells but we know now that the cells of the microenvironment are also key actors of the tumor development and their molecular features need to be addressed.

4) *Can you ensure the Group A is not regions that correspond to infiltrating brain tissue given the process of neurogenesis/synapse function? Group B also sounds very suspicious for necrotic tissue and areas of microvascular proliferation (PEC?).*

In the table presented above, we can see that the three regions are heterogeneous and cannot be directly linked to a unique histologic region annotated by the pathologist. It is true that Group A (red) has more points in the infiltrating tumor areas than the two other groups and Group B (yellow) has more points in the necrotic tissue than the two other groups. Nevertheless, for the three groups, the majority of the points are located in the tumor region.

5) *The survival proteins chosen seem to show only slight differences. The external validation is seen as a positive for the study, but the overall differences are minor given the massive stratification of outcomes (months vs >4 years). It is hard to know what clinical value differences, even if slightly significant at the population level, means for individual patients. For the validation cohort, the patients were classified according to their survival times:*

- Low → less than 1 year

- Intermediate → between 1 and 2 years
- High → between 2 and 5 years.

Based on the expression of the four markers, we can observe that low and intermediate survivors are quite similar. Therefore, we can stratify patients into two groups, the ones who survive less than 2 years and the ones who survive more than 2 years and up to 5 years. Therefore, these markers can help the oncologist to estimate the gravity of the disease and the rapidity of its progression for each individual patient and guide the therapeutic decision. For example, ALCAM was already found higher expressed in glioblastoma in another study (PMID: 22304788). Some therapeutic drugs have been developed targeting ALCAM such as antibody-drug conjugates (PMID: 30962321). Therefore, the proteins we found and linked to the survival of patients could be “druggable”. This could be a major advance for patients with glioblastoma.

We have discussed this point in the discussion (lines 818-825).

Minor

1) Page 8, Line 193 - *Guidec* → *guide*

This was modified.

2) The reference to 96 vs 46 + 50FFPE is confusing. The cohort for the proteomics component was 46 and the additional 50 were for validation/testing. This should be more clearly communicated.

The methods later go one to say only 30 histologically validated samples were used. While the exact number likely doesn't change results, this is the true number used in the analysis and should be highlighted more.

In the materials and methods section, we have described the cohort of patients in the “Patient samples and consent” paragraph. We have added a sentence (line 218) to specify that the 46 prospective tumors were used for proteomics experiments.

The 30 histologically validated samples were used for the spidermass analysis. For mass spectrometry imaging and microproteomics techniques, we have used the 46 tumors (lines 250 and 343). We have clarify this point in the materials and methods section, as well as in the results section (line 472-473). This point is also specified in the abstract.

3) PEC should be more explicitly defined in the manuscript

PEC is the French abbreviation for microvascular proliferation, we have modified the annotations to MVP and explained it in the result section (line 468).

Reviewers' Comments:

Reviewer #1:

Remarks to the Author:

In the revised version of the manuscript by Duhamel et al. entitled "Spatial reference and alternative proteome analysis of glioblastoma reveals molecular signatures and associates survival with specific markers", the authors have properly addressed previous concerns.

Reviewer #2:

Remarks to the Author:

1) The authors have provided more information regarding the overlap of MALDI and pathologist annotations and note there is fairly poor overlap in the majority of cases. This is seen as somewhat concerning especially since they note that MALDI profiling sometimes could not detect the presence of vessels or necrosis. The reviewer sees these areas as "positive controls" and an inability to identify objective and truly "heterogeneous" hallmarks of GBM brings the optimization of the method and usefulness of the data in question. It is difficult to be compelled to believe MALDI can define subtle areas of tumoral heterogeneity if it cannot detect differences in vastly different cell types (glial vs collections of non neoplastic endothelial cells).

2) The tool used in the study is largely a bulk profiling tool which can be highly biased to tissue make up. While I applaud the efforts to quantify the different tissue patterns across the yellow, red and blue clusters the results of this were as expected and make the proteomic differences difficult to provide a fair comparison. In my mind are still likely driving most of the pathways analysis and clustering shown in Figure 2, even if they make up only a non-majority component of the sampled tissue. This is why most bulk profiling studies aim for highly pure tumour regions and, even then, suffer from tissue heterogeneity differences.

2) Without a good correlation with histological hallmarks, the reader is left to believe MALDI is like "black magic", which is OK, but this would require some less supervised approach on the survival analysis to highlight significance. The survival analysis as it is currently done appears highly supervised and can be prone to overfitting of biomarkers. Many of the markers in Fig 3D,E,F overlap bringing concern how sensitivity and specific these are to any biologically relevant aspect of the clusters.

3) The differences in Fig 4 appear very minor despite the large p-values and appear to fluctuate up and down between low, int, high and perhaps only AXNA11 shows some trend across the groupings.

Manuscript NCOMMS-22-15279A

Second Revision

Response of authors to reviewer's comment are in bold

Comments:

Reviewer #1 (Remarks to the Author):

In the revised version of the manuscript by Duhamel et al. entitled "Spatial reference and alternative proteome analysis of glioblastoma reveals molecular signatures and associates survival with specific markers", the authors have properly addressed previous concerns.

Reviewer #2 (Remarks to the Author):

1) The authors have provided more information regarding the overlap of MALDI and pathologist annotations and note there is fairly poor overlap in the majority of cases. This is seen as somewhat concerning especially since they note that MALDI profiling sometimes could not detect the presence of vessels or necrosis. The reviewer sees these areas as "positive controls" and an inability to identify objective and truly "heterogenous" hallmarks of GBM brings the optimization of the method and usefulness of the data in question. It is difficult to be compelled to believe MALDI can define subtle areas of tumoral heterogeneity if it cannot detect differences in vastly different cell types (glial vs collections of non neoplastic endothelial cells).

Response: We have identified that there is still a misunderstanding on these experiments. In the manuscript, we directly present the results of MALDI MS Imaging after the individual and the global segmentation of the image data. We realize that this is confusing. We have therefore revised the manuscript and added data presenting results from the MALDI MS imaging before the multivariate statistical analysis by segmentation (see images below). The images present the distribution of the different ions (m/z) directly extracted from the MS spectra, each corresponding to a different tryptic peptide and thus a protein. In that case it can be clearly seen that we have markers specific to the different areas with distinct histological features which match to what is annotated by the pathologist, considering that the MALDI image and annotations are done from serial sections. Then after we perform the individual segmentation per patient or global for all the patients together, we will search for the main molecular similarities / discriminative features within the tissues. After segmentation there is less correlation with the histological annotation, which is expectable because, for example, all patients have vessels, thus this is not necessarily a discriminative feature between the patients. The segmentation is then showing the main changes due to different types of cell phenotypes and microenvironments, therefore the different proteomic subtypes within the patient cohort for their stratification.

We have added these data (Figure 1B) and revised the manuscript consequently and better explained the segmentation. We agree that this will help understanding and not leave with the feeling that MALDI Imaging is not a robust technology that we obtain opposite results to the initial tissue annotations.

2) The tool used in the study is largely a bulk profiling tool which can be highly biased to tissue make up. While I applaud the efforts to quantify the different tissue patterns across the yellow, red and blue clusters the results of this were as expected and make the proteomic differences difficult to provide a fair comparison. In my mind are still likely driving most of the pathways analysis and clustering shown in Figure 2, even if they make up only a non-majority component of the sampled tissue. This is why most bulk profiling studies aim for highly pure tumour regions and, even then, suffer from tissue heterogeneity differences.

Response: We think that again there is a misunderstanding about the nature of what is obtained from the segmentation data. Again, as shown above, individual ions give access to the tissue molecular heterogeneity and is equivalent to performing immunohistochemistry in multiplex against different markers. On the other hand, the segmentation is not aiming at looking to tumor heterogeneity but finding whether similar molecular features can be found to be common between different patients. Moreover, the segmentation is unsupervised and the clustering of the samples in the large scale

proteomic as well. The fact that certain clusters of proteins are found either up or down regulated in the heatmap for certain group of patients show that these patients have common regulatory pathways. If no features can be found in common between patients, no clusters can be obtained. Besides the pathways related to these clusters of proteins are clearly related to pathological mechanisms. These clusters not driven by MALDI MSI segmentation but formed from statistically unsupervised analysis. This explains why the sample clustering in the heatmap does not always perfectly reflect the MALDI segmentation. For example, blue cluster contains in majority samples from blue segmentation but also a few red and yellow samples. In order to better explain this part, we have revised the text.

3) Without a good correlation with histological hallmarks, the reader is left to believe MALDI is like "black magic", which is OK, but this would require some less supervised approached on the survival analysis to highlight significance. The survival analysis as it is currently done appears highly supervised and can be prone to overfitting of biomarkers. Many of the markers in Fig 3D,E,F overlap bringing concern how sensitivity and specific these are to any biologically relevant aspect of the clusters.

Response: As explained for comment 1, the MALDI imaging data prior to the segmentation confirms the histological hallmarks. As mentioned, we propose to add this data to avoid the reader having the feeling that MALDI is "black magic" giving results that are not correlating with the rest. The survival analysis is not supervised. We have searched if some markers are matching with patient's survival using strong and robust statistical analysis made by a statistician. We also cross-validate these markers with immunohistochemistry, to be sure that, indeed, none of these markers would be a false positive. We have revised the figures (Figures 3E, 4A, 4B, supp Figure 4 and 5) and improve them adding the p value for each of the found marker to show that the variations are significant (****p value <0.0001, *** p value <0.001, ** p value <0.01, * p value <0.05).

3) The differences in Fig 4 appear very minor despite the large p-values and appear to fluctuate up and down between low, int, high and perhaps only AXNA11 shows some trend across the groupings.

Response: Again, we have added data from other markers in Figures 4A and 4B. The additional markers (HSPD1, MAOB and CFH) were selected because they are significantly overexpressed in association with the 2 OS clusters with a $p < 0.05$ (supp. Figure 4). Moreover, they have been previously reported in the literature in association with glioma. For these 10 markers we have provided the boxplots plotted from the fluorescence quantification with the p-value for each marker. The level of expression of the markers are significantly different (see p values) between the survival groups. We have modified Figures 3 and 4 and revised the text to include these new data.

Reviewers' Comments:

Reviewer #2:

Remarks to the Author:

No additional comments. The authors improved the description of their study, which I think will now make it more clear to the readers.